# Sketch to Adapt: Fine-Tunable Sketches for Efficient LLM Adaptation

**Tianyi Zhang** [* 1 2]  **Junda Su** [* 1]  **Aditya Desai** [3]  **Oscar Wu** [2]  **Zhaozhuo Xu** [4]  **Anshumali Shrivastava** [1 2 5 6]

## Abstract

Adapting pre-trained large language models (LLMs) is crucial but challenging due to their enormous size. Parameter-efficient fine-tuning (PEFT) techniques typically employ additive adapters applied to frozen model weights. To further reduce memory usage, model weights are often compressed through quantization. However, existing PEFT methods often yield suboptimal model quality because they rely on restrictive assumptions, such as low-rank constraints on adapters to limit the number of trainable parameters. We find that sketching, a popular data compression technique, can serve as an efficient LLM adaptation strategy while avoiding the low-rank assumption. We introduce SketchTune, a compressive adaptation strategy that compresses LLM weights into compact fine-tunable sketches, integrating compression and adaptation into a unified framework. This integration eliminates the need for complex two-path computation in existing PEFT techniques, enabling faster and more memory-efficient training and inference. SketchTune is supported by mathematical insights into matrix classes that are better approximated using sketching rather than low-rank methods. Our extensive evaluations with Llama and Mistral models demonstrate that SketchTune outperforms leading PEFT methods across diverse tasks while using substantially smaller base models and comparable trainable parameters. As a highlight, SketchTune outperforms LoRA, DoRA, and S$^2$FT on commonsense and math benchmarks using 2.6-3.5$\times$ smaller base models and exceeds LoftQ in accuracy by 14.48% on GSM8K with 7.3$\times$ fewer trainable parameters.

---

[*]Equal contribution [1]Rice University, Houston, TX [2]xMAD.ai [3]University of California, Berkeley, Berkeley, CA [4]Stevens Institute of Technology, Hoboken, NJ [5]ThirdAI Corp. [6]Ken Kennedy Institute. Correspondence to: Tianyi Zhang <tz21@rice.edu>, Anshumali Shrivastava <anshumali@rice.edu>.

*Proceedings of the 42$^{nd}$ International Conference on Machine Learning*, Vancouver, Canada. PMLR 267, 2025. Copyright 2025 by the author(s).

## 1. Introduction

Recent advancements in Large Language Models (LLMs) have demonstrated their potential to drive significant progress in various fields, including natural language processing (Min et al., 2023), reasoning (Wei et al., 2022), and problem-solving (Kojima et al., 2022). These pre-trained LLMs can tackle a wide range of challenges thanks to the extensive knowledge acquired during pre-training, but they still require fine-tuning for optimal performance on specific downstream tasks (Longpre et al., 2023). Unfortunately, fine-tuning LLMs can be prohibitively resource-intensive due to their large size. Many existing works address this issue by adding a small set of additional trainable parameters while fixing the pre-trained parameters (Han et al., 2024).

**Restrictive Linear Algebraic Assumptions in LLM Adapters.** Parameter-efficient fine-tuning (PEFT) methods aim to reduce the number of trainable parameters by imposing specific linear algebraic assumptions on LLM weight updates. For instance, sparsity-based approaches assume that only a small subset of weights undergo updates (Sung et al., 2021; Yang et al., 2024b), while the more popular low-rank adapter-based methods (Hu et al., 2022; Liu et al., 2024b) enforce the restrictive assumption that weight updates are inherently low rank. However, recent studies (Liu et al., 2024a) challenge this assumption, showing that fully fine-tuned weight updates can exhibit high-rank patterns. Our empirical findings further support this, revealing that low-rank representations may not be optimal for capturing weight updates, as illustrated in Figure 1. Additionally, compressing weight updates alone may not be sufficient for fine-tuning LLMs under resource constraints. To address this, existing methods incorporate weight quantization to further lower memory for fine-tuning (Dettmers et al., 2023; Yin et al., 2023; Li et al., 2023b).

**Quantized Fine-Tuning Produces Sub-Optimal Results.** To reduce the memory usage for adapting full LLM parameters, quantized fine-tuning methods freeze the low-bit quantized base weights and update additional low-rank adapters (Dettmers et al., 2023; Yin et al., 2023). However, this combination of the low-rank adapter assumption and quantized weights results in sub-optimal performance compared to PEFT methods using full base models (Li et al., 2023b; Yin et al., 2023). Additionally, since quan-

tized model weights and trainable adapters use different bit widths, input tensors must pass through two unmergeable computation paths during a forward pass, leading to increased latency and lower throughput.

**SketchTune: Fine-Tunable Sketches for Unified Compression and Adaptation.** We introduce SketchTune, a method that unifies compression and adaptation of LLMs with sketching. SketchTune uses a learned sketching algorithm to compress the LLM into a small set of shared sketched parameters. These sketched parameters are fully differentiable, allowing us to directly update them for adaptation. The original parameters can be approximately reconstructed from the shared sketched parameters via a mapping matrix that projects each original parameter to a shared one. By leveraging a carefully designed, learned sketching procedure, SketchTune preserves the pre-trained capabilities of the full model while drastically reducing the model size by 3–8×. Furthermore, SketchTune goes beyond the restrictive low-rank or sparse assumption on weight updates. We provide mathematical insights into the scenarios where weight updates are better approximated by sketching, as well as empirical evidence for why sketching can approximate weight updates with lower errors.

Through extensive experiments, we show that SketchTune models achieve higher average accuracy on commonsense and math reasoning benchmarks compared to competitive PEFT baselines, while utilizing sketched models that are 2.6–3.6× smaller than the full base models used by these baselines. When compared to competitive quantized fine-tuning methods at the 2-bit region, SketchTune achieves 14.48% better accuracy while using 7.4× fewer trainable parameters. By leveraging dedicated CUDA kernels, Sketch-Tune demonstrates better training and inference efficiency than PEFT and quantized fine-tuning methods. Our code and model checkpoints are available publicly[1].

# 2. SketchTune: Fine-Tunable Sketches for LLM Adaptation

Fine-tuning LLMs presents significant computational and memory challenges. To address this, we propose Sketch-Tune, a novel approach that compresses LLM weights into memory-efficient sketches and fine-tunes these sketches directly to adapt to downstream tasks. We begin by motivating the use of sketching for compressing model weight updates through empirical evidence. We then describe our method for learned sketching of LLM weights and the techniques for fine-tuning sketched models efficiently on GPUs.

---

[1] https://github.com/LeanModels/SketchTune

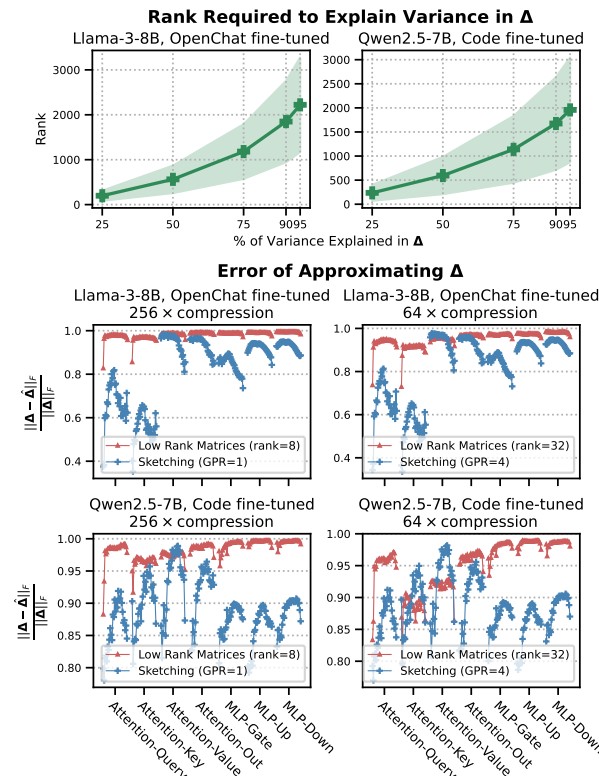

*Figure 1.* (**Top 2**) Minimum rank required by low-rank matrices to explain varying percentages of variance in fine-tuned LLM weight updates. (**Bottom 4**) Optimal approximation errors for sketching and low-rank matrices under different compression ratios.

## 2.1. Weight Updates Are Far from Low-Rank: Sketching Provides a Superior Alternative

In this section, we provide empirical evidence that weight updates resulting from full fine-tuning of LLMs are high-rank, which limits the effectiveness of low-rank approximations for capturing these updates. We further demonstrate that our proposed sketching-based compression technique achieves substantially lower approximation errors than conventional low-rank methods when representing the weight updates.

Let $\mathbf{W}$ represent the original model weights and $\mathbf{W}'$ represent the fine-tuned weights. The weight update, defined as $\boldsymbol{\Delta} \triangleq \mathbf{W}' - \mathbf{W}$, typically requires as much storage as the original weights. LoRA (Hu et al., 2022), a prevalent PEFT method, compresses weight updates by representing them as a product of two low-rank matrices. To evaluate the effectiveness of this approach, we examine the capacity of low-rank matrices to approximate weight updates from two fine-tuned LLMs: Llama-3-8B (Dubey et al., 2024), fine-tuned on OpenChat (Wang et al., 2024), and Qwen2.5-7B, fine-tuned on source code (Yang et al., 2024a).

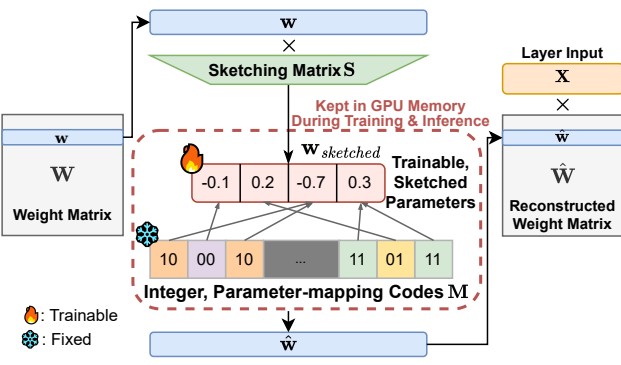

*Figure 2.* An illustration of SketchTune's process of sketching for model compression and fine-tuning.

We perform singular value decomposition (SVD) on each weight update matrix $\mathbf{\Delta}$ to determine the matrix rank required to capture a specified percentage of variance. Figure 1 illustrates the rank (and its standard deviation) necessary to account for different levels of variance in $\mathbf{\Delta}$. The results show that an average rank exceeding 1000 is required to explain merely 75% of the variance. This finding suggests that standard low-rank approaches, typically employing ranks ranging from 4 to 64, may inadequately capture the complexity of fine-tuned weight updates.

In contrast, we propose SketchTune, a novel approach that uses sketching to compress and represent weight updates. Unlike traditional PEFT methods that rely on adapters or additional parameters added to frozen weights, SketchTune directly compresses the entire model through a set of fine-tunable sketched parameters. Specifically, the sketched model utilizes shared parameters along with a mapping matrix that associates each original parameter with these shared parameters.

We compare the quality of SketchTune and low-rank matrices in approximating the weight updates on the previously mentioned Llama and Qwen models. We measure the approximation quality using the normalized approximation error defined as $\frac{\|\mathbf{\Delta}-\hat{\mathbf{\Delta}}\|_F}{\|\mathbf{\Delta}\|_F}$, where $\hat{\mathbf{\Delta}}$ represents the approximated weight update (see Appendix B). Figure 1 shows that SketchTune achieves lower approximation errors than low-rank matrices across most layers, suggesting its superior effectiveness in capturing weight updates.

### 2.2. Formulation: Learning to Sketch LLM Weights

Sketching is a data compression technique that preserves essential properties of the original data while substantially reducing memory requirements. Conventional matrix sketching techniques employ randomized algorithms such as column/row sampling (Liberty, 2013) and random

projection (Raskutti & Mahoney, 2016), but these stochastic methods can corrupt the pre-trained knowledge embedded in LLM weights. To prevent such degradation, we introduce a learned sketching approach that preserves the model's pre-trained capabilities. The core idea is to compress each weight matrix $\mathbf{W} \in \mathbb{R}^{r \times c}$ into a sketched matrix $\mathbf{W}_{sketched} \in \mathbb{R}^{r \times k}$, where $k \ll c$, without significantly affecting the loss of the model. During inference, we recover an approximate weight matrix $\hat{\mathbf{W}} \in \mathbb{R}^{r \times c}$ from $\mathbf{W}_{sketched}$ on-the-fly for matrix multiplications.

For learning sketched matrices, our goal is to keep the loss of the sketched model as close as possible to that of the original network. Let $\mathbf{W}_{\mathcal{N}}$ and $\hat{\mathbf{W}}_{\mathcal{N}}$ denote the tensors representing the weights of network $\mathcal{N}$ and its recovered weights from sketches, respectively. The learning objective for the sketched weights is to minimize $\mathcal{L}(\hat{\mathbf{W}}_{\mathcal{N}})-\mathcal{L}(\mathbf{W}_{\mathcal{N}})$, where $\mathcal{L}(\mathbf{W}_{\mathcal{N}})$ represents the loss evaluated at $\mathbf{W}_{\mathcal{N}}$. To make the learning tractable, we employ an approximation proposed by Nagel et al. (2020):

$$\mathcal{L}(\hat{\mathbf{W}}_{\mathcal{N}}) - \mathcal{L}(\mathbf{W}_{\mathcal{N}}) \approx \sum_{\mathbf{W} \in \mathcal{N}} \|\mathbf{W}\mathbf{X} - \hat{\mathbf{W}}\mathbf{X}\|_F^2 \quad (1)$$

where $\mathbf{W}$ denotes a weight matrix in the network, and $\mathbf{X}$ represents its corresponding input matrix. This approximation offers two key advantages: 1. it decomposes the learning into a layer-wise convex problem, making it computationally feasible, 2. it enables layer-by-layer learning, allowing large models to be processed on a single GPU. Furthermore, since $\|\mathbf{W}\mathbf{X} - \hat{\mathbf{W}}\mathbf{X}\|_F^2$ can be expressed as a sum over the products of all row vectors of $\mathbf{W}$ and $\mathbf{X}$, the learning process can be further decomposed into row-wise independent problems. This leads to the learning objective:

$$\underset{\hat{\mathbf{w}}}{\arg\min} \|\mathbf{w}\mathbf{X} - \hat{\mathbf{w}}\mathbf{X}\|_2^2 \quad (2)$$

where $\mathbf{w}$ and $\hat{\mathbf{w}}$ represent a row in the original weight matrix and its corresponding approximation recovered from row sketches, respectively. We learn the sketched weights by minimizing output distortions introduced by the reconstructed weights, using a small sample drawn from the empirical distribution of $\mathbf{X}$.

### 2.3. A Row-Wise Learning Strategy

For each weight matrix $\mathbf{W} \in \mathbb{R}^{r \times c}$ in the LLM, we independently compress each row vector $\mathbf{w} \in \mathbb{R}^{1 \times c}$ using a learned sketching process. The process involves learning two matrices: a sketching matrix $\mathbf{S}$ for compression and a mapping matrix $\mathbf{M}$ for reconstruction.

The sketching matrix $\mathbf{S} \in \mathbb{R}^{c \times k}$ projects $\mathbf{w}$ into a lower-dimensional space, producing a sketched row $\mathbf{w}_{sketched} \in \mathbb{R}^{1 \times k}$. The mapping matrix $\mathbf{M} \in \{0,1\}^{k \times c}$, a column-wise one-hot binary matrix, recovers an approximation $\hat{\mathbf{w}} \in$

$\mathbb{R}^{1 \times c}$ of the original row for use in matrix multiplications. Formally, this process is defined as:

$$\mathbf{w}_{sketched} = \mathbf{w}\mathbf{S}, \qquad \hat{\mathbf{w}} = \mathbf{w}_{sketched}\mathbf{M}. \qquad (3)$$

After sketching, the original row $\mathbf{w}$ and the sketching matrix $\mathbf{S}$ are no longer required. Only the sketched weights $\mathbf{w}_{sketched}$ and the mapping matrix $\mathbf{M}$ need to be stored in GPU memory during training and inference, leading to resource efficiency.

To learn the mapping matrix $\mathbf{M}$ for parameter reconstruction, we apply the iterative quantization strategy proposed by Frantar et al. (2022b) and Frantar & Alistarh (2022) to preserve model quality. Each column of $\mathbf{M}$ is a binary one-hot vector that maps an original parameter (from a row of size $c$) to one of the $k$ entries in the sketched parameters $\mathbf{w}_{sketched}$, where $k \ll c$. This mapping inevitably introduces some error in the model output. To minimize this error, we learn the columns of $\mathbf{M}$ sequentially and iteratively update the remaining unmapped parameters to compensate for the introduced error after each step. Concretely, for each original parameter, we identify the entry in $\mathbf{w}_{sketched}$ that is closest to it and assign the corresponding one-hot value to $\mathbf{M}$. After fixing this column of $\mathbf{M}$, we apply an update $\boldsymbol{\delta}$ (Equation 13 in the Appendix) to the remaining unmapped parameters to absorb the approximation error. This process is repeated until all columns of $\mathbf{M}$ are assigned.

For learning the sketched parameters $\mathbf{w}_{sketched}$, a straightforward method is to perform clustering on the $c$ original parameters to obtain a set of $k$ centers as the sketched parameters. To minimize the impact on model quality, we prioritize preserving the precision of parameters with large, outlying inverse Hessian diagonals $\frac{1}{\mathbf{H}_{i,i}^{-1}}$, where $\mathbf{H}$ is the second-order derivative of Equation 2. We achieve this by adopting the learning objective proposed by Zhang & Shrivastava, which emphasizes the preservation of more influential parameters:

$$\underset{\mathbf{w}_{sketched} \in \mathbb{R}^k}{\arg\min} \sum_i \left(\frac{1}{\mathbf{H}_{i,i}^{-1}}\right)^s \left| \text{RTN}(\mathbf{w}_i, \mathbf{w}_{sketched}) - \mathbf{w}_i \right|^2 \qquad (4)$$

where $\text{RTN}(\mathbf{w}_i, \mathbf{w}_{sketched})$ (round-to-nearest operator) rounds the value of $\mathbf{w}_i$ to its nearest sketched parameter in $\mathbf{w}_{sketched}$, and $s$ is a hyperparameter controlling the emphasis on preserving outliers of $\frac{1}{\mathbf{H}_{i,i}^{-1}}$, which we set $s = 3$. More details on the mathematical derivations are presented in Appendix D.

Using Equation 4 as objective, we learn $\mathbf{w}_{sketched}$ by leveraging the weighted k-means (Lloyd, 1982; Zhang & Amini, 2021) algorithm. Consequently, we learn the sketching matrix $\mathbf{S}$ as follows. Let $a_1, \dots, a_c \in \{1, \dots, k\}$ be the cluster indices of parameters $\mathbf{w}_1, \dots, \mathbf{w}_c$ weighted by

$\left(\frac{1}{\mathbf{H}_{1,1}^{-1}}\right)^s, \dots, \left(\frac{1}{\mathbf{H}_{c,c}^{-1}}\right)^s$, produced by the weighted k-means algorithm. Then, the $i$-th row and $j$-th column of the sketching matrix $\mathbf{S}$ is given as follows:

$$\mathbf{S}_{i,j} = \begin{cases} \dfrac{\left(\frac{1}{\mathbf{H}_{i,i}^{-1}}\right)^s \mathbf{w}_i}{\sum_l \left(\frac{1}{\mathbf{H}_{l,l}^{-1}}\right)^s} & \text{if } a_i = j \\ 0 & \text{otherwise} \end{cases} \qquad (5)$$

The final learning procedure for model sketching proceeds as follows: we first learn the sketching matrix $\mathbf{S}$ through weighted k-means, and obtain the sketched parameters $\mathbf{w}_{sketched}$ as $\mathbf{w}\mathbf{S}$. We then initialize the mapping matrix $\mathbf{M}$ to be empty, and iteratively learn the columns of $\mathbf{M}$. In each learning step, we fix the next column of $\mathbf{M}$ to map the next parameter to its nearest sketched parameter in $\mathbf{w}_{sketched}$, and apply an update to the unmapped parameters to compensate for the errors. During the iterative process, the error in the weight update can accumulate, degrading sketched model quality. To address this, we apply the strategy proposed by Frantar et al. (2022b) to use the Cholesky reformulation for inverse Hessian calculations and apply the weight updates in a block-wise manner. Specifically, we divide the weights into blocks of $B = 128$ columns and keep weight updates contained to those columns. Once all parameters within the block have been mapped, we apply a global weight update to the rest of the unmapped parameters.

### 2.4. Scaling Up Number of Trainable Parameters

To scale up the learning capacity of the sketched models, we need to increase the count of trainable parameters. Unlike adapter-based methods, the number of trainable parameters in SketchTune models is fixed after model sketching. Hence, to allow more flexibility in the amount of sketched parameters, we propose to divide each row into multiple sub-rows, and sketch each sub-row independently. Specifically, we divide each row $\mathbf{w} \in \mathbb{R}^{1 \times c}$ into $g$ non-overlapping, contiguous groups of sub-rows $\mathbf{w}' \in \mathbb{R}^{1 \times \frac{c}{g}}$. With $g$ groups per row, we are increasing the number of trainable parameters $g$-fold compared to row-wise sketching. We use the notation SketchTune$_{\text{GPR}=g}$ to represent a sketched model with $g$ groups per row (GPR). With everything put together, the final algorithm for weight sketching is given in Algorithm 1. We present a quality comparison between the sketched models and the original models in Appendix G.

### 2.5. Fine-Tuning Sketches

Once the model weights have been sketched, the original weights $\mathbf{w}$ and the sketching matrix $\mathbf{S}$ are no longer needed for training or inference. During training and inference, we use the sketched weights $\mathbf{w}_{sketched}$ and the mapping matrix $\mathbf{M}$ to reconstruct weights $\hat{\mathbf{w}}$ as $\mathbf{w}_{sketched}\mathbf{M}$. Thus, with $\mathbf{X}$ being the layer input, the forward pass computes

**Algorithm 1** Learning to Sketch LLM Weights

1: **Function** LearnSketchingMatrix($\mathbf{w}, \mathbf{X}$)
  **Input:** sub-row weights $\mathbf{w}'$, layer input $\mathbf{X}$
  **Output:** sketching matrix $\mathbf{S}$
2: $a_1, \ldots, a_c \leftarrow$ WeightedKMeansCluster(
   $[\mathbf{w}'_1, \ldots, \mathbf{w}'_{\frac{c}{G}}], [(\frac{1}{\mathbf{H}_{1,1}^{-1}})^s, \ldots, (\frac{1}{\mathbf{H}_{\frac{c}{G}, \frac{c}{G}}^{-1}})^s], k)$
3: let $\mathbf{S} \in \mathbb{R}^{\frac{c}{G}, k}$, $\mathbf{S}_{i,j} \leftarrow \begin{cases} \dfrac{\left(\frac{1}{\mathbf{H}_{i,i}^{-1}}\right)^s \mathbf{w}_i}{\sum_l \left(\frac{1}{\mathbf{H}_{l,l}^{-1}}\right)^s} & \text{if } a_i = j \\ 0 & \text{otherwise} \end{cases}$
4: **return** $\mathbf{S}$

5: **Function** LearnToSketch($\mathbf{w}, \mathbf{X}, G$)
  **Input:** row weights $\mathbf{w}$, layer input $\mathbf{X}$, groups per row $G$
  **Output:** sketched weights of all $G$ group $w_{sketched}^0, \ldots, w_{sketched}^{G-1}$, the mapping matrix $\mathbf{M}$
6: $\mathbf{M} \leftarrow \mathbf{0}^{k \times c}$ ▷ initialize mapping matrix
7: $\hat{\mathbf{w}} \leftarrow \mathbf{0}^{1 \times c}$ ▷ initialize reconstructed weights
8: $\mathbf{e} \leftarrow \mathbf{0}^{1 \times B}$ ▷ initialize weight errors
9: $\mathbf{H}^{-1} \leftarrow$ cholesky$\left([2\mathbf{X}\mathbf{X}^\top]^{-1}\right)$
10: **for** $g \leftarrow 0, \ldots, G - 1$ **do**
11: $\quad \mathbf{w}' \leftarrow \mathbf{w}_{g\frac{c}{G}:(g+1)\frac{c}{G}}$ ▷ get current sub-row
12: $\quad \mathbf{S} \leftarrow$ LearnSketchingMatrix($\mathbf{w}', \mathbf{X}$)
13: $\quad \mathbf{w}_{sketched}^g \leftarrow \mathbf{w}'\mathbf{S}$
14: $\quad$ **for** $i \leftarrow g\frac{c}{G}, g\frac{c}{G} + B, g\frac{c}{G} + 2B, \ldots, (g+1)\frac{c}{G}$ **do**
15: $\quad\quad$ **for** $j \leftarrow i, \ldots, i + B - 1$ **do**
16: $\quad\quad\quad m \leftarrow \arg\min_l \left| [\mathbf{w}_{sketched}^g]_l - \mathbf{w}_j \right|$
17: $\quad\quad\quad \mathbf{M}_{m,j} \leftarrow 1$ ▷ set the current mapping column
18: $\quad\quad\quad \hat{\mathbf{w}}_j \leftarrow [\mathbf{w}_{sketched}^g]_m$
19: $\quad\quad\quad \mathbf{e}_{j-i} \leftarrow \dfrac{\mathbf{w}_j - \hat{\mathbf{w}}_j}{\mathbf{H}_{j,j}^{-1}}$
20: $\quad\quad\quad \mathbf{w}_{j:(i+B)} \leftarrow \mathbf{w}_{j:(i+B)} - \mathbf{e}_{j-i}\mathbf{H}_{j,j:(i+B)}^{-1}$
21: $\quad\quad$ **end for**
22: $\quad\quad \mathbf{w}_{(i+B):} \leftarrow \mathbf{w}_{(i+B):} - \mathbf{e}\mathbf{H}_{i:(i+B),(i+B):}^{-1}$
23: $\quad$ **end for**
24: **end for**
25: **return** $[w_{sketched}^0, \ldots, w_{sketched}^{G-1}], \mathbf{M}$

$\mathbf{y} = \mathbf{w}_{sketched}\mathbf{M}\mathbf{X}$. For adaptation, we freeze $\mathbf{M}$ and perform back-propagation to update the sketched parameters $\mathbf{w}_{sketched}$. The gradients of the sketched parameters are given as:

$$\frac{\partial \mathcal{L}}{\partial \mathbf{w}_{sketched}} = \frac{\partial \mathcal{L}}{\partial \mathbf{y}}(\mathbf{M}\mathbf{X})^\top \qquad (6)$$

We present an illustration of the sketching and fine-tuning process of SketchTune in Figure 2.

## 2.6. Custom CUDA Kernel for Efficient Training and Inference

We develop dedicated CUDA kernels for efficient training and inference of sketched models on GPUs by leveraging the shared memory (see Appendix E for details). In Section 4.2, we perform a comprehensive evaluation of the efficiency of SketchTune during training and inference and compare it against competitive methods. To ensure efficient storage, we store the mapping matrix $\mathbf{M} \in \{0, 1\}^{k \times c}$, a column-wise one-hot binary matrix, as an integer matrix. Due to its one-hot nature, each column of $\mathbf{M}$ can be compactly represented with the index of its one-hot entry using a $\lceil \log_2 k \rceil$-bit integer. To leverage the full bit widths, we take $k \in \{16, 8, 4\}$ to use the data types INT4, INT3, and INT2, respectively.

## 3. Theoretical Analysis

The properties of the true update matrix $\boldsymbol{\Delta}$, i.e. the update matrix obtained after full fine-tuning, determine a good assumption for compression of weight update. However, the true $\boldsymbol{\Delta}$ is not known apriori. In this section, we analyze characteristics of $\boldsymbol{\Delta}$ and the effect of sketching-based methods such as SketchTune, especially against popular low-rank approximation alternatives. Since $\boldsymbol{\Delta}$ and the mappings $\mathbf{M}$s for each row derived from $\mathbf{W}$ can be unrelated, it is safe to assume $\{\mathbf{M}\}$ is random w.r.t $\boldsymbol{\Delta}$. For ease of exposition, we assume that $\{\mathbf{M}\}$s belong to a specific kind of random sketching matrices derived from random-fold hashing (Desai & Shrivastava, 2023). Our result is presented below:

**Theorem 3.1.** *Consider a matrix $\boldsymbol{\Delta} : n \times n$ with sorted (descending) singular values $\{\rho_i\}_{i=1}^n$, squares of which are drawn from power law $i^{-\eta}$ parameterized by coefficient $\eta$. Under the compression factor $\alpha$ (i.e. using $n^2/\alpha$ parameters), let low-rank approximation and sketch approximation be $\boldsymbol{\Delta}_l$ and $\boldsymbol{\Delta}_s$ respectively. Then, the low-rank error is*

$$||\boldsymbol{\Delta} - \boldsymbol{\Delta}_l||_F^2 = ||\boldsymbol{\Delta}||_F^2 - \sum_{i=1}^{n/2k} \rho_i^2 \qquad (7)$$

*The expected error of random-fold sketching approximation is,*

$$\mathbf{E}(||\boldsymbol{\Delta} - \boldsymbol{\Delta}_l||_F^2) = ||\boldsymbol{\Delta}||_F^2 - \frac{1}{\alpha}\left(\sum_{i=1}^n \rho_i^2\right) \qquad (8)$$

*For large enough $n$, the expected sketching approximation error is smaller than low-rank approximation error if*

$$\eta \in \left[0, 1 - \frac{\log(\alpha)}{\log(2\alpha)}\right] \qquad (9)$$

The proof of the theorem is presented in Appendix F. The above theorem characterizes matrices that are well approximated by sketching instead of low-rank decomposition. It

*Table 1.* Accuracy of SketchTune compared to competitive PEFT methods for fine-tuning Llama models on math datasets. Baseline results are taken from Yang et al. (2024b). SketchTune achieves better or comparable accuracy while using sketched models that are 2.6–3.6× smaller than the full base models used by other PEFT methods.

| Model | Method | Base Model (GB) | Trainable Param (M) | MultiArith | GSM8K | AddSub | AQuA | SingleEq | SVAMP | MAWPS | Avg. ↑ |
|---|---|---|---|---|---|---|---|---|---|---|---|
| GPT-3.5 | - | - | - | 83.8 | 56.4 | 85.3 | 38.9 | 88.1 | 69.9 | 87.4 | 72.8 |
| | Full FT | 13.48 | 6,738.4 | 98.8 | 43.1 | 91.1 | 20.9 | 94.3 | 60.6 | 88.2 | 71.0 |
| | LoRA | 13.48 | 55.9 | 98.0 | 40.0 | 91.2 | 21.7 | 93.1 | 56.7 | 85.3 | 69.7 |
| | DoRA | 13.48 | 56.6 | 97.3 | 38.9 | 89.6 | 22.4 | 93.9 | 58.4 | 85.3 | 69.4 |
| LLaMA-7B | S$^2$FT | 13.48 | 54.6 | **98.8** | **41.3** | 91.4 | 21.3 | 93.5 | **58.4** | 86.1 | 70.1 |
| | SketchTune$_{GPR=1}$ | 3.89 | 21.8 | 97.8 | 36.5 | 89.9 | **25.2** | 90.7 | 55.7 | 86.6 | 68.9 |
| | SketchTune$_{GPR=2}$ | 3.93 | 43.5 | 96.8 | 39.0 | **92.2** | 20.1 | 92.7 | 55.5 | 86.6 | 69.0 |
| | SketchTune$_{GPR=4}$ | 4.02 | 87.0 | 98.3 | 39.7 | 90.9 | 22.0 | 93.5 | 58.0 | 87.4 | 70.0 |
| | SketchTune$_{GPR=8}$ | 4.19 | 174.1 | 98.3 | 40.6 | 91.9 | 19.7 | **95.1** | 57.5 | **88.7** | **70.3** |
| | Full FT | 26.03 | 13,015.9 | 98.3 | 47.6 | 92.9 | 26.0 | 95.1 | 65.7 | 88.7 | 73.5 |
| | LoRA | 26.03 | 87.2 | 97.5 | 47.8 | 89.9 | 20.5 | 94.3 | 61.2 | 87.4 | 71.2 |
| | DoRA | 26.03 | 88.5 | 97.2 | 48.1 | 90.6 | 20.9 | 93.9 | 63.8 | 88.2 | 71.8 |
| LLaMA-13B | S$^2$FT | 26.03 | 84.6 | 97.7 | **48.4** | 90.4 | 22.8 | 95.5 | 63.9 | 87.8 | 72.4 |
| | SketchTune$_{GPR=1}$ | 7.14 | 34.1 | 97.2 | 44.0 | 88.6 | 26.0 | 91.7 | 64.9 | 85.7 | 71.2 |
| | SketchTune$_{GPR=2}$ | 7.21 | 68.2 | 98.2 | 46.9 | 91.1 | **27.2** | 93.9 | 61.8 | 86.6 | 72.2 |
| | SketchTune$_{GPR=4}$ | 7.36 | 136.3 | 98.5 | 47.8 | 91.9 | 24.0 | **95.9** | 64.2 | **89.1** | 73.1 |
| | SketchTune$_{GPR=8}$ | 7.67 | 272.6 | **98.8** | 47.6 | **92.2** | 25.2 | 95.5 | **66.8** | 87.4 | **73.4** |
| | Full FT | 13.48 | 6,738.4 | 99.3 | 47.5 | 91.1 | 24.4 | 96.7 | 62.5 | 89.1 | 72.9 |
| | LoRA | 13.48 | 55.9 | 97.5 | 44.0 | 91.2 | 20.9 | 94.1 | 59.2 | 85.7 | 70.4 |
| | DoRA | 13.48 | 56.6 | 98.2 | 43.8 | 90.1 | 24.4 | 94.5 | 59.1 | 89.1 | 71.3 |
| LLaMA2-7B | S$^2$FT | 13.48 | 54.6 | 98.5 | 44.3 | 91.1 | 25.2 | 94.7 | **61.8** | 88.2 | 72.0 |
| | SketchTune$_{GPR=1}$ | 3.92 | 21.8 | 98.0 | 41.4 | 89.6 | **26.4** | 92.9 | 59.3 | 89.1 | 71.0 |
| | SketchTune$_{GPR=2}$ | 3.97 | 43.5 | 98.8 | 43.5 | 92.2 | 20.5 | 95.3 | 59.9 | 89.1 | 71.3 |
| | SketchTune$_{GPR=4}$ | 4.05 | 87.0 | **99.3** | 46.5 | 91.1 | 23.2 | 94.5 | 59.8 | 88.2 | 71.8 |
| | SketchTune$_{GPR=8}$ | 4.23 | 174.1 | 98.7 | 46.5 | **93.9** | 24.0 | **96.7** | 61.7 | **90.3** | **73.1** |
| | Full FT | 16.06 | 8,030.3 | 99.2 | 62.0 | 93.9 | 26.8 | 96.7 | 74.0 | 91.2 | 77.7 |
| | LoRA | 16.06 | 56.2 | 99.5 | 61.6 | 92.7 | 25.6 | 96.3 | 73.8 | 90.8 | 77.2 |
| | DoRA | 16.06 | 57.0 | 98.8 | 62.7 | 92.2 | 26.8 | 96.9 | 74.0 | 91.2 | 77.5 |
| LLaMA3-8B | S$^2$FT | 16.06 | 56.2 | 99.7 | 65.8 | **93.7** | **31.5** | 97.8 | 76.0 | 92.4 | 79.6 |
| | SketchTune$_{GPR=1}$ | 5.77 | 22.0 | 97.8 | 66.3 | 90.1 | 26.8 | 95.5 | **79.8** | 90.8 | 78.2 |
| | SketchTune$_{GPR=2}$ | 5.81 | 44.0 | 98.3 | **69.4** | 90.6 | 29.5 | 94.3 | 76.8 | 91.2 | 78.6 |
| | SketchTune$_{GPR=4}$ | 5.92 | 88.1 | 99.2 | 68.2 | 91.4 | 30.7 | 97.0 | 76.2 | 92.4 | 79.3 |
| | SketchTune$_{GPR=8}$ | 6.10 | 176.2 | **99.7** | 68.8 | 92.7 | 29.1 | **98.6** | 77.5 | **92.9** | **79.9** |

implies if the update-matrix $\Delta$ is close to full-rank, i.e. $\eta$ is closer to 0, then SketchTune is well suited to approximate $\Delta$, whereas $\eta$ closer to 1 would make low-rank a superior alternative. Clearly, as we can see from Figure 1, that $\Delta$ is far from being low rank, which indicates the superiority of SketchTune over low-rank approximations.

## 4. Experiments

We conduct comprehensive experiments to evaluate the adaptation capabilities of SketchTune against competitive baselines across diverse tasks, including math problem solving, commonsense reasoning, and instruction following. Below, we introduce the datasets, models, and baselines used for evaluation, as well as the software and hardware for conducting the experiments.

**Models and Benchmarks.** We fine-tune and evaluate the following models: 1. Llama-7B, 2. Llama-13B (Touvron et al., 2023a), 3. Llama-2-7B, 4. Llama-2-13B (Touvron et al., 2023b), 5. Llama-3-8B (Dubey et al., 2024), 6. Mistral-7B (Jiang et al., 2023). For math problem-solving, we fine-tune models on the Math10K dataset and evaluate on 7 different math reasoning datasets (Hu et al., 2023). For commonsense reasoning, we fine-tune on the Commonsense170K dataset and evaluate on 8 different commonsense reasoning datasets (Hu et al., 2023). For instruction fine-tuning, we fine-tune Mistral-7B (Jiang et al., 2023) on the Alpaca-GPT4 dataset (Peng et al., 2023) for one epoch and evaluate it on MT-Bench (Zheng et al., 2023) using GPT-4o as a judge. Detailed descriptions of the datasets are provided in Appendix H. To compare SketchTune against efficient quantized model fine-tuning methods, we follow the settings in Li et al. (2023b) to fine-tune and test Llama-2 models

*Table 2.* Accuracy of SketchTune compared to competitive PEFT methods for fine-tuning Llama models on commonsense reasoning datasets. Baseline results are taken from Yang et al. (2024b). SketchTune achieves better or comparable accuracy while using sketched models that are 2.7–3.5× smaller than the full base models used by other PEFT methods.

| Model | Method | Base Model (GB) | Trainable Param (M) | BoolQ | PIQA | SIQA | HellaSwag | Wino | ARC-e | ARC-c | OBQA | Avg.↑ |
|---|---|---|---|---|---|---|---|---|---|---|---|---|
| ChatGPT | - | - | - | 73.1 | 85.4 | 68.5 | 78.5 | 66.1 | 89.8 | 79.9 | 74.8 | 77.0 |
| Llama-7B | Full FT | 13.48 | 6,738.4 | 70.3 | 84.2 | 80.1 | 92.3 | 85.4 | 86.6 | 72.8 | 83.4 | 81.9 |
| | LoRA | 13.48 | 55.9 | 69.2 | 81.7 | 78.4 | 83.4 | 80.8 | 79.0 | 62.4 | 78.4 | 76.7 |
| | DoRA | 13.48 | 56.6 | 68.5 | 82.9 | 79.6 | 84.8 | 80.8 | 81.4 | 65.8 | 81.0 | 78.1 |
| | Galore | 13.48 | 55.9 | 68.6 | 79.0 | 78.5 | 84.7 | 80.1 | 80.3 | 62.1 | 77.3 | 76.3 |
| | LoReFT | 13.48 | 2.0 | 69.3 | 84.4 | **80.3** | 93.1 | 84.2 | 83.2 | 68.2 | 78.9 | 80.2 |
| | LISA | 13.48 | 667.8 | 70.4 | 82.1 | 78.7 | 92.4 | 82.9 | 84.9 | 70.2 | 78.4 | 80.0 |
| | S$^2$FT | 13.48 | 54.6 | **72.7** | 83.7 | 79.6 | 93.4 | 83.5 | 86.1 | **72.2** | 83.4 | 81.8 |
| | SketchTune$_{GPR=4}$ | 4.02 | 87.0 | 72.1 | **85.6** | 80.2 | **93.7** | **84.6** | **86.2** | 71.0 | **84.8** | **82.3** |
| Llama-13B | Full FT | 26.03 | 13,015.9 | 74.5 | 86.3 | 81.3 | 94.4 | 86.9 | 89.7 | 77.9 | 88.8 | 85.0 |
| | LoRA | 26.03 | 87.2 | 72.1 | 83.5 | 80.5 | 90.5 | 83.7 | 82.8 | 68.3 | 82.4 | 80.5 |
| | DoRA | 26.03 | 88.5 | 72.4 | 84.9 | 81.5 | 92.4 | 84.2 | 84.2 | 69.6 | 82.8 | 81.5 |
| | LoReFT | 26.03 | 3.9 | 72.1 | 86.3 | 81.8 | 95.1 | **87.2** | 86.2 | 73.7 | 84.2 | 83.3 |
| | S$^2$FT | 26.03 | 84.6 | **74.2** | 85.7 | 80.7 | 94.9 | 86.4 | 88.4 | **76.3** | 87.8 | 84.3 |
| | SketchTune$_{GPR=4}$ | 7.36 | 136.3 | 73.9 | **87.4** | **82.5** | **95.6** | 86.1 | **90.3** | 75.7 | **89.4** | **85.1** |
| Llama-2-7B | Full FT | 13.48 | 6,738.4 | 74.7 | 84.9 | 78.7 | 93.7 | 84.1 | 87.5 | 75.2 | 85.0 | 83.0 |
| | LoRA | 13.48 | 55.9 | 69.8 | 79.9 | 79.5 | 83.6 | 82.6 | 79.8 | 64.7 | 81.0 | 77.6 |
| | DoRA | 13.48 | 56.6 | 71.8 | 83.7 | 76.0 | 89.1 | 82.6 | 83.7 | 68.2 | 82.4 | 79.7 |
| | S$^2$FT | 13.48 | 54.6 | 72.9 | 86.1 | 80.2 | **94.3** | **85.5** | 87.2 | 74.6 | 83.4 | 83.0 |
| | SketchTune$_{GPR=4}$ | 4.05 | 87.0 | **73.3** | 86.2 | 81.2 | 94.1 | 85.4 | **87.6** | 75.2 | **85.8** | **83.6** |
| Llama-3-8B | Full FT | 16.06 | 8,030.3 | 73.9 | 86.2 | 79.1 | 93.1 | 85.8 | 88.1 | 78.2 | 84.0 | 83.6 |
| | LoRA | 16.06 | 56.2 | 70.8 | 85.2 | 79.7 | 92.5 | 84.9 | 88.9 | 78.7 | 84.4 | 82.5 |
| | DoRA | 16.06 | 57.0 | 74.6 | 89.3 | 79.9 | 95.5 | 85.6 | 90.5 | 80.4 | 85.8 | 85.2 |
| | S$^2$FT | 16.06 | 56.2 | 75.0 | 89.0 | 80.7 | **96.5** | 88.0 | 92.5 | **83.4** | 87.8 | 86.6 |
| | SketchTune$_{GPR=4}$ | 5.92 | 88.1 | **75.0** | **90.2** | **82.7** | 95.9 | **88.2** | **92.6** | 82.1 | **89.4** | **87.0** |

*Table 3.* MT-Bench scores for Mistral-7B fine-tuned on the Alpaca-GPT4 training set. The baseline results are taken from Yang et al. (2024b). Despite using a 3.1× smaller base model, SketchTune achieves a better average score than baselines.

| Method | Base Model (GB) | Writing | Roleplay | Reasoning | Code | Math | Extraction | STEM | Humanities | Avg. |
|---|---|---|---|---|---|---|---|---|---|---|
| Full FT | 14.48 | 5.50 | 4.45 | 5.45 | 2.50 | 3.25 | 5.78 | 4.75 | 5.45 | 4.64 |
| S$^2$FT | 14.48 | **6.95** | 4.40 | 5.50 | 2.70 | 3.55 | 5.95 | 6.35 | 6.75 | 5.27 |
| SketchTune$_{GPR=4}$ | 4.66 | 4.60 | **5.20** | 9.23 | 3.05 | 4.80 | 7.45 | 8.13 | 8.45 | **6.36** |

on the language modeling dataset WikiText-2 (Merity et al., 2022) and the math reasoning dataset GSM8K (Cobbe et al., 2021). Additional instruction following evaluations against quantized model fine-tuning methods are presented in Appendix K.

**Baselines.** We compare SketchTune against the following PEFT baselines: 1. Galore (Zhao et al., 2024), 2. LoReFT (Wu et al., 2024), 3. LISA (Pan et al., 2024), 4. LoRA (Hu et al., 2022), 5. DoRA (Liu et al., 2024b), 6. S$^2$FT (Yang et al., 2024b). We also report the results of full fine-tuning, GPT-3.5 (text-Davinci-003), and ChatGPT (gpt-3.5-turbo) from Hu et al. (2023). For Sketch-Tune, we use sketched models compressed with the INT4 data type and compare them against baselines that use the original weights as the base model. For baseline meth-

ods that fine-tune quantized models, we use 1. QLoRA (Dettmers et al., 2023), 2. LoftQ (Li et al., 2023b) and compare with SketchTune at different bit widths. We optimize SketchTune's hyper-parameters, including learning rate and batch size, through a parameter sweep, and we report the hyper-parameters for training in Appendix I.

**Software and Hardware.** We implement SketchTune using PyTorch (Paszke et al., 2019) and the Transformers library (Wolf et al., 2020). We develop custom CUDA kernels optimized for the specific operations required in SketchTune, ensuring high-performance execution on modern GPU architectures. We sketch each model using a single Quadro RTX 8000-48GB GPU. For model training, we train each model using a single NVIDIA A100-40GB GPU.

*Table 4.* Perplexity and accuracy of SketchTune compared to QLoRA and LoftQ, two efficient fine-tuning methods for quantized models, at various bit-widths. Baseline results are taken from Li et al. (2023b). N.A denotes that the model failed to converge. SketchTune achieves better or comparable perplexity and accuracy while using 1.8–29.4× less trainable parameters than baseline methods.

| Method | Data Type | WikiText-2 | | | | Method | Data Type | GSM8K | | | |
| | | Llama-2-7B | | Llama-2-13B | | | | Llama-2-7B | | Llama-2-13B | |
| | | Trainable Param (M) | PPL ↓ | Trainable Param (M) | PPL ↓ | | | Trainable Param (M) | ACC↑ | Trainable Param (M) | ACC↑ |
|---|---|---|---|---|---|---|---|---|---|---|---|
| LoRA$_{rank=64}$ | FP16 | 159.91 | 5.08 | 250.35 | 5.12 | LoRA$_{rank=64}$ | FP16 | 159.91 | 36.90 | 250.35 | 43.10 |
| LoRA$_{rank=64}$+Reg | FP16 | 159.91 | - | 250.35 | - | LoRA$_{rank=64}$+Reg | FP16 | 159.91 | 34.40 | 250.35 | 45.30 |
| QLoRA$_{rank=64}$ | NF4 | 159.91 | 5.70 | 250.35 | 5.22 | QLoRA$_{rank=64}$ | NF4 | 159.91 | 35.10 | 250.35 | 39.90 |
| LoftQ$_{rank=64}$ | NF4 | 159.91 | **5.24** | 250.35 | 5.16 | LoftQ$_{rank=64}$ | NF4 | 159.91 | 35.00 | 250.35 | 45.00 |
| SketchTune$_{GPR=1}$ | INT4 | 21.76 | 5.32 | 34.08 | **4.81** | SketchTune$_{GPR=4}$ | INT4 | 87.03 | **39.73** | 136.31 | **50.34** |
| QLoRA$_{rank=64}$ | NF3 | 159.91 | 5.73 | 250.35 | 5.22 | QLoRA$_{rank=64}$ | NF3 | 159.91 | 32.10 | 250.35 | 40.70 |
| LoftQ$_{rank=64}$ | NF3 | 159.91 | 5.63 | 250.35 | 5.13 | LoftQ$_{rank=64}$ | NF3 | 159.91 | 32.90 | 250.35 | 44.40 |
| SketchTune$_{GPR=1}$ | INT3 | 10.90 | **5.63** | 17.04 | **5.05** | SketchTune$_{GPR=4}$ | INT3 | 43.50 | **37.15** | 68.16 | **47.54** |
| QLoRA$_{rank=64}$ | NF2 | 159.91 | N.A | 250.35 | N.A. | QLoRA$_{rank=64}$ | NF2 | 159.91 | N.A. | 250.35 | N.A. |
| LoftQ$_{rank=64}$ | NF2 | 159.91 | 7.85 | 250.35 | 7.69 | LoftQ$_{rank=64}$ | NF2 | 159.91 | 20.90 | 250.35 | 25.40 |
| SketchTune$_{GPR=1}$ | INT2 | 5.44 | **7.40** | 8.52 | **6.22** | SketchTune$_{GPR=4}$ | INT2 | 21.75 | **29.95** | 34.08 | **39.88** |

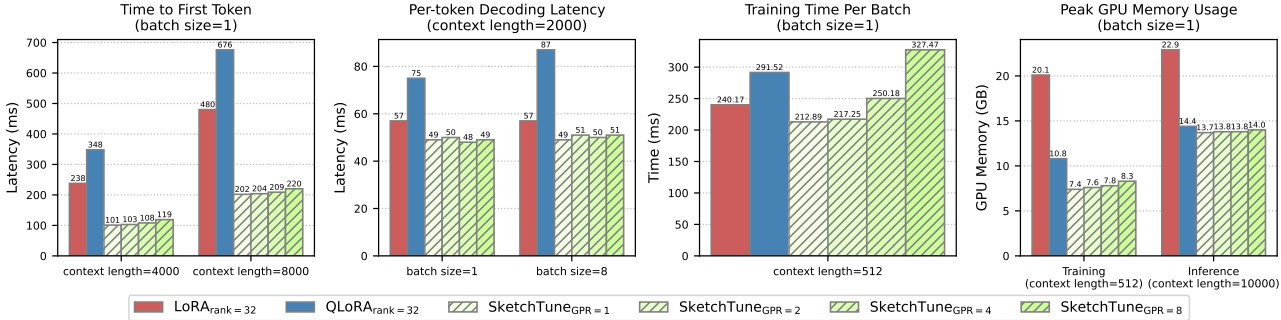

*Figure 3.* A comparison on the training and inference efficiency of SketchTune, LoRA, and QLoRA.

## 4.1. Results

### 4.1.1. COMPARISON WITH PEFT METHODS

**Math Problem-Solving.** Table 1 presents the accuracy results of models fine-tuned on the Math10K dataset (Hu et al., 2023). SketchTune models are compressed to use INT4 representation for weights, while baselines use full models for fine-tuning. We explore the effectiveness of SketchTune at different compression rates by varying groups per row (GPR) in the sketched models. We highlight that SketchTune$_{GPR=8}$ consistently achieves the best accuracy compared to PEFT baselines despite using 2.6-3.4× smaller models, and is on par with full fine-tuning and GPT-3.5. Regarding parameter efficiency, SketchTune$_{GPR=2}$ performs better or on par with LoRA and DoRA, despite using fewer trainable parameters. We also conclude that scaling up the number of parameters by increasing GPR leads to a consistent improvement in the average accuracy across datasets.

**Commonsense Reasoning.** Table 2 presents the accuracy results of models fine-tuned on the Commonsense170K dataset. To maintain a practical trade-off between memory

efficiency and final performance, all SketchTune models use the INT4 representation and GPR=4 for model sketching, while the baseline methods employ the original weights as the base. Despite using 2.7-3.5× smaller base model size, SketchTune consistently outperforms competitive PEFT methods and even full fine-tuning in the average accuracy across benchmarks.

**Instruction Fine-tuning** Table 3 shows the MT-Bench scores, judged by GPT-4o, for SketchTune, S$^2$FT, and fully fine-tuned models. SketchTune outperforms both baselines on most tasks, while using 67.8% smaller base model.

### 4.1.2. COMPARISON WITH COMPRESSIVE FINE-TUNING

Table 4 reports the perplexity on WikiText-2 and accuracy on GSM8K for models fine-tuned with QLoRA, LoftQ, and SketchTune at different bit widths. QLoRA and LoftQ use 4/3/2-bit NormalFloat (NF4/NF3/NF2) (Dettmers et al., 2023), while SketchTune uses 4/3/2-bit integers (INT4/INT3/INT2). SketchTune achieves lower perplexity on WikiText-2 in most cases and higher accuracy on GSM8K across all bit widths. Notably, it outperforms LoftQ

by 4.73 and 5.34 points on GSM8K (4-bit) with $1.8\times$ fewer trainable parameters and by 9.05 and 14.48 points (2-bit) with $7.3\times$ fewer parameters. Moreover, SketchTune surpasses LoftQ on WikiText-2 (2-bit) while using $29.4\times$ fewer trainable parameters. Details on calculating SketchTune's number of trainable parameters can be found in Appendix C.

### 4.2. Efficiency Analysis

In Figure 3, we report the training and inference efficiency of SketchTune. Moreover we compare SketchTune with LoRA and QLoRA. All experiments are performed on an NVIDIA A100-40GB GPU. The results are averaged over 10 runs, each with 10 warmup steps. LoRA and QLoRA-based methods keep base and adapter weights separately, which requires two matrix multiplications for each layer, leading to inefficiencies and difficulties in implementation. SketchTune is free of adapters and requires a single matrix multiplication for each layer. SketchTune demonstrates $2.0$-$2.4\times$ lower time to first token (TTFT) than LoRA, and $2.9$-$3.3\times$ lower TTFT than QLoRA. For decoding latency, SketchTune is consistently faster than LoRA and QLoRA. For training time, SketchTune is lower or on par with baselines. Regarding GPU memory usage, SketchTune consumes $1.6$-$2.7\times$ less memory than LoRA during training and inference due to the smaller size of sketched models.

## 5. Related Works

**Resource-Efficient Fine-Tuning of LLMs.** As fine-tuning LLMs is resource intensive, existing works reduce the computational and memory resources demands through parameter-efficient adapters, optimizer state compression, base model quantization, and more (Han et al., 2024). Adapter-based methods attempt to reduce the parameters for capturing model weight updates through low-rank methods (Hu et al., 2022; Liu et al., 2024b), vector-based random matrix adaptation (Kopiczko et al., 2024), sparsity (Guo et al., 2021; Yang et al., 2024b), orthogonal fine-tuning (Qiu et al., 2023; Liu et al., 2024c), etc. Optimizer states typically consume twice the amount of memory as trainable parameters (Loshchilov & Hutter, 2019), and existing works reduce this overhead through low-rank approximation (Zhao et al., 2024) and quantization (Dettmers et al., 2022; Li et al., 2023a). To reduce the memory demand for the base model, existing works (Dettmers et al., 2023; Li et al., 2023b; Qin et al., 2024; Yin et al., 2023) quantize the full model to integers and fine-tune full-precision adapters added on top. To make LLMs more memory efficient, existing works apply compression to the model weights (Frantar et al., 2022a; Zhang et al., 2025), activations (Xiao et al., 2023), and KV cache (Zhang et al., 2024a;b), while maintaining model accuracy.

**Sketching for Model Compression.** Sketching techniques have been explored as effective methods for compressing neural networks (Xu, 2025), aiming to reduce computational and storage requirements while maintaining performance. Random sketching for model compression has been explored recently in a variety of settings, including embedding compression (Chen et al., 2015; Desai et al., 2022; Desai & Shrivastava, 2022) and general model compression (Desai et al., 2023; Desai & Shrivastava, 2023). However, these methods are not suitable for compressing models after training; they are most effective for training compressed models from scratch. SketchTune is among the first to explore sketching techniques for post-training model compression. Other techniques include multi-hashing (Eban et al., 2020), tensor sketching (Kasiviswanathan et al., 2017), random projection (Ravi, 2019), linear sketches (Daniely et al., 2016), and higher-order count sketch (Shi & Anandkumar, 2020).

## 6. Conclusion

In this work, we introduced SketchTune, a novel approach that unifies model compression and adaptation through weight sketching. Our method addresses the fundamental limitations of existing PEFT approaches by eliminating low-rank constraints and avoiding the computational overhead of separate adapter paths. Through theoretical analysis and comprehensive empirical evaluation across diverse tasks, we demonstrated that SketchTune achieves superior performance while using significantly smaller base models than competitive baselines. These results establish SketchTune as a promising direction for efficient adaptation of LLMs.

## Acknowledgements

This work was supported by National Science Foundation SHF-2211815 and Ken Kennedy Institute Cluster Grants.

## Impact Statement

This paper presents work whose goal is to advance the field of Machine Learning. There are many potential societal consequences of our work, none which we feel must be specifically highlighted here.

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

# Appendix

## A. Mathematical Notations

A summary of the mathematical notations used in the paper is presented in Table 5.

*Table 5.* Notations used in the paper.

| Notation | Type | Explanations |
|---|---|---|
| $r, c$ | $\mathbb{Z}^+, \mathbb{Z}^+$ | The number of rows and columns in a weight matrix |
| $k$ | $\mathbb{Z}^+$ | The number of compressed columns in sketches |
| $\mathbf{W}, \mathbf{w}$ | $\mathbb{R}^{r \times c}, \mathbb{R}^{1 \times c}$ | A weight matrix, a row of the weight matrix |
| $\mathbf{W}_{sketched}, \mathbf{w}_{sketched}$ | $\mathbb{R}^{r \times k}, \mathbb{R}^{1 \times k}$ | The sketched parameters of a weight matrix, the sketched parameters of a row |
| $\hat{\mathbf{W}}, \hat{\mathbf{w}}$ | $\mathbb{R}^{r \times c}, \mathbb{R}^{1 \times c}$ | The reconstructed weight matrix and row from sketched parameters |
| $\mathbf{S}$ | $\mathbb{R}^{c \times k}$ | The sketching matrix for compression, where $\mathbf{wS} = \mathbf{w}_{sketched}$ |
| $\mathbf{M}$ | $\{0, 1\}^{k \times c}$ | The mapping matrix for reconstruction, where $\hat{\mathbf{w}} = \mathbf{w}_{sketched}\mathbf{M}$ |

## B. Calculating Errors of Weight Update Approximation

In this section, we describe the details for calculating the approximation errors of weight updates using low-rank matrices and sketching. We use the update $\mathbf{\Delta}$ of two fully fine-tuned models: 1. `openchat/openchat-3.5-1210` (Wang et al., 2024) from the base model `meta-llama/Meta-Llama-3-8B` (Dubey et al., 2024), 2. `Qwen/Qwen2.5-Coder-7B` from the base model `Qwen/Qwen2.5-7B` (Yang et al., 2024a).

We use the metric of normalized approximation error $\frac{\|\mathbf{\Delta} - \hat{\mathbf{\Delta}}\|_F}{\|\mathbf{\Delta}\|_F}$, where $\hat{\mathbf{\Delta}}$ is the best approximation of $\mathbf{\Delta}$ achievable using low-rank matrices or sketching. For low-rank matrices, we calculate $\hat{\mathbf{\Delta}}$ using the first $r$ dominant entries of singular value decomposition (SVD), where $r$ is the rank of the low rank matrices. For sketching, we first sketch the models using the SketchTune algorithm to derive the row-wise mapping matrix $\mathbf{M}$. Then, we calculate a row of the optimal approximation $\hat{\boldsymbol{\delta}}$ as

$$\hat{\boldsymbol{\delta}} = \mathbf{M}(\mathbf{M}^\top \mathbf{M})^{-1} \mathbf{M}^\top \boldsymbol{\delta} \tag{10}$$

## C. Calculating Total Number of Sketched Parameters

In this section, we detail the method for calculating the number of trainable parameters in a SketchTune model. Our approach involves sketching all linear projection layers within a LLM, excluding the token embedding layer and the final prediction head layer.

Each row of the weight matrix in these projection layers is divided into $g$ sketching groups, where $g$ represents the Groups Per Row (GPR) configured prior to sketching. Utilizing an $n$-bit weight representation, each parameter within a sketching group is encoded using an $n$-bit integer, allowing for $2^n$ distinct values. Consequently, the number of trainable parameters for each linear projection layer is calculated as:

$$\text{Trainable Parameters} = \text{Number of Rows} \times g \times 2^n$$

To illustrate, consider sketching the LLaMA-2-7B model with GPR = 4 and INT4 weight representation. Each transformer layer in the LLaMA-2-7B model includes key, query, value, and output projection layers with dimensions $4096 \times 4096$, as well as Multi-Layer Perceptron (MLP) layers sized $4096 \times 11008$ (down), $11008 \times 4096$ (up), and $11008 \times 4096$ (gate).

For all 32 transformer layers, the total number of trainable parameters is:

$$\text{Total Trainable Parameters} = 32 \times 4 \times 2^4 \times (4096 \times 4 + 4096 + 11008 \times 2) = 87,031,808$$

Given that the full model comprises 6,738 million parameters, this results in a compression ratio of approximately $77\times$.

## D. Details of Row-Wise Sketch Learning

We denote the weight reconstruction error $\boldsymbol{\delta}$ and the loss error $\epsilon$, where

$$\boldsymbol{\delta} \triangleq \hat{\mathbf{w}} - \mathbf{w}, \qquad \epsilon \triangleq \mathcal{L}(\mathbf{w} + \boldsymbol{\delta}) - \mathcal{L}(\mathbf{w}). \tag{11}$$

Our objective is to solve for the optimal weight update $\boldsymbol{\delta}$ to apply to the weights $\mathbf{w}$ such that the loss error $\epsilon$ is minimized. The learning process then proceeds as follows: we sequentially learn and fix the $i$-th column of the mapping matrix, denoted as $\mathbf{M}_{:,i}$, during the $i$-th step. At each step, based on the loss error $\epsilon_i$ introduced by the current mapping, we solve for the optimal weight update $\boldsymbol{\delta}_i$ that minimizes $\epsilon_i$, and apply the update to the unmapped original weights. This process is repeated until all columns of $\mathbf{M}$ have been fixed.

To simplify solving for $\boldsymbol{\delta}$, we use a second-order Taylor expansion to approximate the loss error $\epsilon$ (Nagel et al., 2020):

$$\epsilon \approx (\frac{\partial \mathcal{L}}{\partial \mathbf{w}})^\top \boldsymbol{\delta} + \frac{1}{2}\boldsymbol{\delta}^\top \mathbf{H}\boldsymbol{\delta}, \quad \text{where } \mathbf{H} = \frac{\partial^2 \mathcal{L}}{\partial \mathbf{w}^2},$$
$$\approx \frac{1}{2}\boldsymbol{\delta}^\top \mathbf{H}\boldsymbol{\delta}, \quad \text{since } \frac{\partial \mathcal{L}}{\partial \mathbf{w}} \approx \mathbf{0} \text{ in a pre-trained LLM.} \tag{12}$$

For the Hessian $\mathbf{H}$, we approximate it by leveraging the second-order derivative of Equation 2, where $\mathbf{H} = 2\mathbf{X}\mathbf{X}^\top$. In practice, we compute $\mathbf{X}$ using a small calibration dataset consisting of 128 sequences of 2048 tokens sampled from the C4 dataset (Dodge et al., 2021). Equation 12 can now be solved with a Lagrangian (Frantar & Alistarh, 2022), which yields the following solutions for $\boldsymbol{\delta}_i$ and $\epsilon_i$:

$$\boldsymbol{\delta}_i = \frac{\mathbf{w}_{sketched}\mathbf{M}_{:,i} - \mathbf{w}_i}{\mathbf{H}_{i,i}^{-1}}\mathbf{H}_{:,i}^{-1}, \epsilon_i = \frac{1}{2}\frac{(\mathbf{w}_{sketched}\mathbf{M}_{:,i} - \mathbf{w}_i)^2}{\mathbf{H}_{i,i}^{-1}}. \tag{13}$$

where $\mathbf{H}_{i,i}^{-1}, \mathbf{H}_{:,i}^{-1}$, is the $i$-th diagonal entry, and the $i$-th column of the Hessian, respectively.

From the solution of $\epsilon_i$, we gain two key insights towards minimizing the loss error:

$$\epsilon_i \propto (\mathbf{w}_{sketched}\mathbf{M}_{:,i} - \mathbf{w}_i)^2, \qquad \epsilon_i \propto \frac{1}{\mathbf{H}_{i,i}^{-1}}. \tag{14}$$

These two facts lead to two takeaways: 1. Since the loss error is proportional to the squared difference between the sketched parameter and the original parameter, a column of $\mathbf{M}$ should always be set to map an original parameter to its nearest sketched parameter. 2. As the loss error is proportional to the inverse Hessian diagonals, the sketched parameters should be optimized to prioritize preserving the precision of weights with larger inverse Hessian diagonals. Therefore, to learn the set of $k$ sketched parameters, we perform clustering to derive $k$ centers from the $c$ original parameters. We further leverage the learning objective proposed by Zhang & Shrivastava to emphasize preserving parameters with outlier inverse Hessian diagonals:

$$\underset{\mathbf{w}_{sketched} \in \mathbb{R}^k}{\arg\min} \sum_i \left(\frac{1}{\mathbf{H}_{i,i}^{-1}}\right)^s \left| \text{RTN}(\mathbf{w}_i, \mathbf{w}_{sketched}) - \mathbf{w}_i \right|^2, \tag{15}$$

where $\text{RTN}(\mathbf{w}_i, \mathbf{w}_{sketched})$ (round-to-nearest operator) maps the parameter $\mathbf{w}_i$ to the nearest sketched parameter in $\mathbf{w}_{sketched}$, and $s$ is a hyperparameter controlling the emphasis on preserving outliers in the inverse Hessian diagonals. In our experiments, we set $s = 3$.

## E. CUDA Kernel Implementation Details

We develop custom CUDA kernels for efficient training and inference of SketchTune. Specifically, we develop two dedicated CUDA kernels: 1. A kernel for *weight reconstruction*, which computes $\hat{\mathbf{W}}$ from $\mathbf{W}_{sketched}$. 2. A kernel for *gradient computation* of the sketched parameters, which calculates $\frac{\partial \mathbf{W}_{sketched}}{\partial \mathcal{L}}$ from $\frac{\partial \hat{\mathbf{W}}}{\partial \mathcal{L}}$.

During training, the approximate weights $\hat{\mathbf{W}}$ are only reconstructed when needed and not kept in memory to save memory usage. For the weight reconstruction kernel, each thread block is responsible for reconstructing a single row of weights by computing $\hat{\mathbf{w}} = \mathbf{w}_{sketched}\mathbf{M}$. The mapping matrix $\mathbf{M}$ is stored in a packed integer format. Each thread block allocates

sufficient shared memory to cache $\mathbf{w}_{sketched}$ (utilizing only 32 bytes for $k = 16$). It then reads the integer indices from $\mathbf{M}$ to perform low-latency lookups from $\mathbf{w}_{sketched}$ in shared memory and writes the retrieved entries of $\hat{\mathbf{w}}$ to global memory.

For the gradient computation kernel, each thread block handles the computation of the gradient for a single row of sketched parameters, specifically calculating

$$\frac{\partial \mathcal{L}}{\partial \mathbf{w}_{sketched}} = \frac{\partial \mathcal{L}}{\partial \hat{\mathbf{w}}} \mathbf{M}^\top.$$

Each thread block allocates enough memory to cache $\frac{\partial \mathcal{L}}{\partial \mathbf{w}_{sketched}}$ $t$ times, where $t$ is the number of threads in each thread block. Threads within the same thread block reads different entries of $\frac{\partial \mathcal{L}}{\partial \hat{\mathbf{w}}}$ and accumulates the values into its own copy of $\frac{\partial \mathcal{L}}{\partial \mathbf{w}_{sketched}}$. Since each thread maintains its own accumulator, atomic operations are unnecessary for ensuring consistency. Finally, an aggregation step combines the intermediate results from all threads to produce the final gradient.

## F. Theory

For the sake of analysis, consider a square matrix $\hat{\mathbf{W}} : n \times n$ of post-quantized weights, where $c = r = n$. Let the true update under full-scale fine-tuning be $\Delta$. Let $\Delta_l$ be the low-rank approximation learned via Low-rank methods. Let $\Delta_s$ be the sketch-based adaptation learned via methods such as SketchTune. Let the compression factor be $\alpha$. For simplicity, we assume that $\alpha \in \mathbb{N}$ and $\alpha | n$.

**Errors with low-rank approximation**    Under the learning, the best approximation of the $\Delta$ under low-rank methods can be given by Eckart-Young-Mirsky theorem,

$$||\Delta - \Delta_l||_F^2 = \sum_{i=n/2k+1}^{n} \rho_i^2 = ||\Delta||_F^2 - \sum_{i=1}^{n/2k} \rho_i^2 \tag{16}$$

where $\rho_i$ is the $i^{th}$ largest singular value.

**Errors with SketchTune**    Consider a single row of the matrix $\mathbf{w} : 1 \times n$, the sketched matrix $\mathbf{w}_{sketched} : 1 \times k$, the mapping discovered $\mathbf{M} : k \times n$. Consider how SketchTune approximates the true corresponding row in $\Delta$, say $\delta$. It tries to learn the best possible solution inside the **row-subspace**($\mathbf{M}$). To analyze the errors, we will make the following assumptions,

- For ease of exposition, we assume the mapping $\mathbf{M}$ is a balanced mapping where each parameter in $\mathbf{w}_{sketched}$ is used equal number of times.

- Since $\delta$ is not known apriori, we assume that mapping $\mathbf{M}$ is a random w.r.t $\delta$ with the distribution over $\mathbf{M}$ being the random-fold mapping defined in (Desai & Shrivastava, 2023).

Under these assumptions, we can analyze the error incurred by SketchTune while approximating $\Delta_i$. The load of each parameter in $\mathbf{w}_{sketched}$ is $\alpha = n/k$. The approximation error can be written as follows.

$$||\delta_s - \delta||_2^2 = \sum_i \left( \delta_i - g(i) \frac{\sum_{j, h(i)=h(j)} g(j)\delta_j}{\alpha} \right)^2 \tag{17}$$

$$\mathbf{E}\left(||\delta_s - \delta||_2^2\right) = \sum_i \left( \frac{(\alpha-1)^2}{\alpha^2} \delta_i^2 + \frac{1}{\alpha^2} \sum_{j \neq i, h(j)=h(i)} \delta_j^2 \right) \tag{18}$$

The $\delta_j$ appears in $\alpha - 1$ other terms. So aggregating,

$$\mathbf{E}\left(||\delta_s - \delta||_2^2\right) = \sum_i \left( \frac{(\alpha-1)^2}{\alpha^2} \delta_i^2 + \frac{\alpha-1}{\alpha^2} \delta_i^2 \right) \tag{19}$$

$$\mathbf{E}\left(||\delta_s - \delta||_2^2\right) = \sum_i \left(\frac{(\alpha - 1)}{\alpha}\delta_i^2\right) \tag{20}$$

$$\mathbf{E}\left(||\delta_s - \delta||_2^2\right) = \frac{(\alpha - 1)}{\alpha}||\delta||_2^2 \tag{21}$$

Considering all the rows together and linearity of expectations,

$$\mathbf{E}\left(||\Delta_s - \Delta||_2^2\right) = \frac{(\alpha - 1)}{\alpha}||\Delta||_2^2 \tag{22}$$

It is important to note that depending on the $\Delta$, one of the approximations will be better than the other. For instance, if $\Delta$ is indeed low-rank, then LoRA will be the best approximation to use. We will show that if $\Delta$ is near full rank, then Sketching dominates LoRA-based approximations. We quantify this observation below,

Let the squared singular values, indexed by $i$ and sorted in the descending order, be represented by power-law $i^{-\eta}$ parameterized by coefficient $\eta$. If $\eta = 0$, then all the singular values are 1, and as $\eta$ increases, the $\Delta$ becomes more low-rank. $\eta = 1$ implies logarithmic sparsity. So we assume $\eta \in [0, 1)$

$$\rho_i^2 = \rho^2(i) = i^{-\eta} \tag{23}$$

Since $\rho_i$ is monotonically decreasing, it can be bounded as follows,

$$\int_{x=1}^{n+1} \rho^2(x)dx \le \sum_{i=1}^{n} \rho_i^2 \le 1 + \int_{x=1}^{n} \rho^2(x)dx \tag{24}$$

$$\left[\frac{x^{1-\eta}}{1-\eta}\right]_1^{n+1} < \sum_{i=1}^{n} \rho_i^2 < 1 + \left[\frac{x^{1-\eta}}{1-\eta}\right]_1^{n} \tag{25}$$

$$\frac{1}{1-\eta}\left((n+1)^{1-\eta} - 1\right) < \sum_{i=1}^{n} \rho_i^2 < 1 + \frac{1}{1-\eta}\left(n^{1-\eta} - 1\right) \tag{26}$$

$$L(n) = \frac{1}{1-\eta}\left((n+1)^{1-\eta} - 1\right) < \sum_{i=1}^{n} \rho_i^2 < 1 + \frac{1}{1-\eta}\left(n^{1-\eta} - 1\right) = R(n) \tag{27}$$

Let us quantify the $\eta$ for which Sketching dominates Low-rank. Under the budget $k$ for each row, $\alpha = n/k$. Sketching is superior in expectation if,

$$\mathbf{E}\left(||\Delta_s - \Delta||_F^2\right) < ||\Delta_l - \Delta||_F^2 \tag{28}$$

which requires,

$$\frac{1}{\alpha}\left(\sum_{i=1}^{n} \rho_i^2\right) > \sum_{i=1}^{n/2\alpha} \rho_i^2 \tag{29}$$

Using the bounds defined above,

$$\frac{1}{\alpha}L(n) > R(n/2\alpha) \tag{30}$$

For large $n$, we can approximate the ratio as

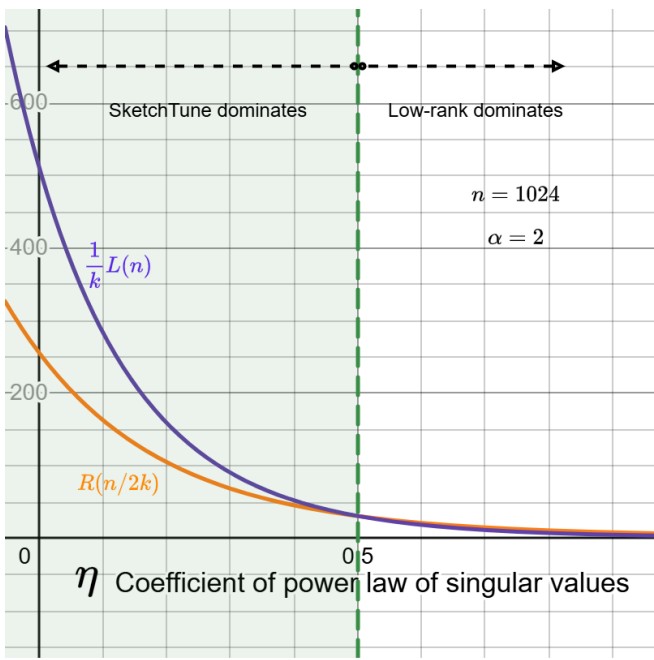

*Figure 4.* Depending on the power-law coefficient in the singular values (a soft proxy for rank) of the unknown $\Delta$ matrix, one method dominates the other. We show this classification for a sample case of $n = 1024$ and $k = 2$

$$\frac{\frac{1}{\alpha}L(n)}{R(n/2\alpha)} = \frac{1}{\alpha}\frac{\frac{1}{1-\eta}\left((n+1)^{1-\eta}-1\right)}{1+\frac{1}{1-\eta}\left(\frac{n^{1-\eta}}{(2\alpha)^{1-\eta}}-1\right)} \tag{31}$$

$$\frac{\frac{1}{\alpha}L(n)}{R(n/2\alpha)} = \frac{1}{\alpha}\frac{\frac{1}{1-\eta}}{\frac{1}{1-\eta}\frac{1}{(2\alpha)^{1-\eta}}} \tag{32}$$

$$\frac{\frac{1}{\alpha}L(n)}{R(n/2\alpha)} = \frac{1}{\alpha}\frac{1}{\frac{1}{(2\alpha)^{1-\eta}}} \tag{33}$$

This fraction is equal to 1 when

$$\alpha = (2\alpha)^{1-\eta} \tag{34}$$

Thus,

$$\eta = 1 - \frac{\log(\alpha)}{\log(2\alpha)} \tag{35}$$

Thus Sketching dominates the low-rank approximation on average in the region $\eta \in \left[0, 1 - \frac{\log(\alpha)}{\log(2\alpha)}\right]$. An example of the bounds is shown in figure 4

**Theorem F.1.** *Consider a matrix $\Delta : n \times n$ with sorted (descending) singular values $\{\rho_i\}_{i=1}^n$, squares of which are drawn from power law $i^{-\eta}$ parameterized by coefficient $\eta$. Under the compression factor $\alpha$ (i.e. using $n^2/\alpha$ parameters), let low-rank approximation and sketch approximation be $\Delta_l$ and $\Delta_s$ respectively. Then, the low-rank error is*

$$||\Delta - \Delta_l||_F^2 = ||\Delta||_F^2 - \sum_{i=1}^{n/2k}\rho_i^2 \tag{36}$$

*The expected error of random-fold sketching approximation is ,*

$$\mathbf{E}(||\Delta - \Delta_l||_F^2) = ||\Delta||_F^2 - \frac{1}{\alpha}\left(\sum_{i=1}^{n}\rho_i^2\right) \tag{37}$$

*For large enough $n$, the expected sketching approximation error is smaller than the low-rank approximation error if*

$$\eta \in \left[0, 1 - \frac{\log(\alpha)}{\log(2\alpha)}\right] \tag{38}$$

## G. Model Sketching Details

*Table 6.* Perplexity of SketchTune, without fine-tuning, on WikiText-2, PTB, and C4 compared to full model. We also report the overhead of sketching time.

| Model | Data Type | GPR | Model Size (GB) | Sketching Time (minutes) | WikiText-2 ↓ | PTB ↓ | C4 ↓ |
|---|---|---|---|---|---|---|---|
| | FP16 | - | - | - | 5.68 | 41.15 | 7.34 |
| | INT2 | 4 | 3.89 | 34.93 | 12.37 | 199.47 | 15.10 |
| | INT3 | 4 | 3.93 | 37.86 | 6.29 | 48.92 | 8.11 |
| Llama-7B | INT4 | 1 | 3.89 | 35.03 | 5.82 | 44.22 | 7.53 |
| | INT4 | 2 | 3.93 | 40.06 | 5.81 | 43.49 | 7.51 |
| | INT4 | 4 | 4.02 | 49.58 | 5.82 | 43.44 | 7.51 |
| | INT4 | 8 | 4.19 | 106.28 | 5.77 | 43.12 | 7.50 |
| | FP16 | - | - | - | 5.09 | 28.10 | 6.80 |
| | INT2 | 4 | 7.14 | 60.58 | 9.23 | 63.68 | 11.08 |
| | INT3 | 4 | 7.21 | 64.18 | 5.55 | 30.15 | 7.30 |
| Llama-13B | INT4 | 1 | 7.14 | 62.70 | 5.20 | 27.67 | 6.91 |
| | INT4 | 2 | 7.21 | 71.28 | 5.20 | 28.34 | 6.91 |
| | INT4 | 4 | 7.36 | 88.97 | 5.18 | 28.51 | 6.90 |
| | INT4 | 8 | 7.67 | 112.43 | 5.20 | 28.04 | 6.90 |
| | FP16 | - | - | - | 5.47 | 37.91 | 7.26 |
| | INT2 | 4 | 3.92 | 35.20 | 15.91 | 166.08 | 18.48 |
| | INT3 | 4 | 3.97 | 38.13 | 6.14 | 42.46 | 8.13 |
| Llama-2-7B | INT4 | 1 | 3.92 | 33.24 | 5.67 | 52.39 | 7.46 |
| | INT4 | 2 | 3.97 | 38.91 | 5.62 | 47.09 | 7.45 |
| | INT4 | 4 | 4.05 | 48.10 | 5.61 | 43.07 | 7.43 |
| | INT4 | 8 | 4.23 | 66.91 | 5.62 | 38.93 | 7.44 |
| | BF16 | - | - | - | 6.14 | 11.18 | 9.45 |
| | INT2 | 4 | 5.77 | 74.63 | 28.52 | 36.04 | 30.79 |
| | INT3 | 4 | 5.81 | 78.83 | 7.73 | 12.87 | 12.09 |
| Llama-3-8B | INT4 | 1 | 5.77 | 69.56 | 6.59 | 11.59 | 10.18 |
| | INT4 | 2 | 5.81 | 79.00 | 6.54 | 11.63 | 10.12 |
| | INT4 | 4 | 5.92 | 88.83 | 6.52 | 11.52 | 10.09 |
| | INT4 | 8 | 6.10 | 106.28 | 6.47 | 11.49 | 10.00 |

We performed model sketching using C4 (Dodge et al., 2021) as calibration dataset (for computing the Hessian $\mathbf{H}$ for model sketching), consisting of 128 sample sequences each with 2048 tokens.

All model are sketched using a single Nvidia Quadro RTX8000 GPU. In Table 6, we provide details on model sketching overhead (sketching time), as well as perplexity comparisons against the original models on WikiText-2 (Merity et al., 2022), PTB (Marcus et al., 1993), and C4 (Dodge et al., 2021) dataset, using 128 sequences of 2048 tokens each.

While SketchTune introduces an additional sketching step before fine-tuning, this preprocessing is fast, resource-efficient, and one-time per base model. In Table 7, we report additional end-to-end sketching time and memory usage (INT4, GPR=4) for different sized models, using a single A100-40GB GPU and the aforementioned calibration setup.

*Table 7.* End to end sketching time (using INT4, GPR=4 setup) for different sized models on a single A100-40GB GPU. Thanks to our layer-wise optimization objective (equation (1)), the sketching process scales efficiently to large models (70B).

| Model | Original Size (GB) | Sketched Size (GB) | Max GPU Mem (GB) | Sketching Time (min) |
|---|---|---|---|---|
| Llama-3.2-3B | 6.43 | 3.18 | 9.92 | 20.70 |
| Llama-3.1-8B | 16.07 | 5.92 | 18.37 | 41.62 |
| Llama-3.1-70B | 141.12 | 40.15 | 28.05 | 266.87 |

## H. Dataset Information

**Math Problem-Solving** To fine-tune and evaluate on math problem solving tasks, we fine-tuned our models on the Math10K dataset (Hu et al., 2023), which includes the training set from GSM8K (Cobbe et al., 2021), AQuA (Ling et al., 2017), and MAWPS (Koncel-Kedziorski et al., 2016) and agumented with language model generated chain-of-thoughts steps. We performed evaluation on 7 math datasets: MultiArith (Roy & Roth, 2016), GSM8K (Cobbe et al., 2021), AddSub (Hosseini et al., 2014), AQuA (Ling et al., 2017), SingleEQ (Koncel-Kedziorski et al., 2015), SVAMP (Patel et al., 2021), and MAWPS (Koncel-Kedziorski et al., 2016). For each test sample, the model performs generation. And a final answer is extracted to calculate model's response accuracy.

**Commonsense Reasoning** The commonsense reasoning tasks consists of questions from 8 different datasets: BoolQ (Clark et al., 2019), PIQA (Bisk et al., 2020), SIQA (Sap et al., 2019), HellaSwag (Zellers et al., 2019), WinoGrande (Sakaguchi et al., 2019), Arc-e, Arc-c (Clark et al., 2018), and OBQA (Mihaylov et al., 2018). The training set consists of training data from all 8 datasets, formatted using a consistent pre-defined template (Hu et al., 2023), resulting in 170K samples. The test set from each dataset is then used individually to evaluate the fine-tuned model's performance.

**WikiText-2** The WikiText-2 dataset (Merity et al., 2016) consists of 44.8k training data, consisting of 36.7K training data, 3.76K validatiaon data, and 4.36K test data. Following LoftQ (Li et al., 2023b), we used the training set to perform fine-tuning, and the validataion set to evaluate fine-tuned model's performance.

**MT-Bench** The MT-Bench dataset (Zheng et al., 2023) is a set of 80 challenging multi-turn open-ended questions across 8 categories: writing, humanities, STEM, extraction, coding, math, reasoning, and roleplay.

**Alpaca** We used the Alpaca dataset (Taori et al., 2023) to evaluate SketchTune's performance on language generation tasks. The Alpaca dataset consist of 52K training samples, with no test or validation split. The responses are generated with the text-davinci-003 engine. Alpaca-GPT4 (Peng et al., 2023) is a similar dataset, which contains outputs generated by GPT-4 using the same prompts as the original Alpaca dataset. We adopted the FastChat framework from Zheng et al. (2023) to perform pair-wise compitition, using GPT-4o as a judge for model generation quality.

## I. Experimental Settings

In this section, we provide a comprehensive overview of the training and evaluation settings employed in our experiments and benchmarks. We begin by detailing the hyperparameter configurations used for each experiment. Following this, we describe the training and generation settings utilized to profile the training and inference efficiency of SketchTune.

### I.1. Hyperparameter Selection

I.1.1. COMPARISON WITH PEFT METHODS

We followed experimental settings described in S$^2$FT (Yang et al., 2024b). We used AdamW (Loshchilov & Hutter, 2019) optimizer for all our experiments. The optimal hyperparamters chosen to produce the final results are provided in Table 8.

I.1.2. COMPARISON WITH COMPRESSIVE FINE-TUNING

For comparing with QLoRA and LoftQ, we followed the experimental settings described in LoftQ (Li et al., 2023b). We used AdamW (Loshchilov & Hutter, 2019) optimizer for all our experiments. The optimal hyperparameters chosen to

*Table 8.* Hyperparameter selections for fine-tuning SketchTune on math reasoning and commonsense reasoning tasks.

| Task | Model | LR | Optimizer | Batch Size | Epochs | LR Scheduler | Warmup Steps |
|---|---|---|---|---|---|---|---|
| Math Reasoning | Llama-7B Llama-13B Llama-2-7B | $8\times10^{-5}$ | AdamW | 16 | 4 | linear | 100 |
| | Llama-3-8B | $3\times10^{-5}$ | AdamW | 16 | 4 | linear | 100 |
| Commonsense Reasoning | Llama-7B Llama-13B Llama-2-7B | $8\times10^{-5}$ | AdamW | 64 | 2 | linear | 100 |
| | Llama-3-8B | $2\times10^{-5}$ | AdamW | 64 | 2 | linear | 100 |

produce the final results are provided in Table 9.

*Table 9.* Hyperparameter selections for fine-tuning SketchTune on WikiText-2 and GSM8K tasks.

| Task | Model | LR | Optimizer | Batch Size | Epochs | LR Scheduler |
|---|---|---|---|---|---|---|
| WikiText-2 | Llama-2-7B Llama-2-13B | $3\times10^{-5}$ | AdamW | 4 | 4 | cosine |
| GSM8k | Llama-2-7B Llama-2-13B | $8\times10^{-5}$ | AdamW | 16 | 4 | cosine |

### I.1.3. INSTRUCTION FINE-TUNING

For instruction fine-tuning tasks on Mistral-7B, we trained on the Alpaca-GPT4 dataset for one epoch. We employed a learning rate of $8 \times 10^{-6}$, AdamW (Loshchilov & Hutter, 2019) for optimizer, linear LR scheduler, and a batch size of 16 with 100 warmup steps.

### I.2. Efficiency Evaluation Settings

Details on memory and speed efficiency evaluation settings are provided in Table 10.

*Table 10.* Efficiency evaluation settings for inference and training

| Stage | Metric | Context Length | Batch Size | Warmup |
|---|---|---|---|---|
| Inference | Time to First Token | 4000 | 1 | 10 |
| | | 8000 | 1 | 10 |
| | Decoding Latency | 2000 | 1 | 10 |
| | | 2000 | 8 | 10 |
| | Peak Memory | 10000 | 1 | 10 |
| Training | Training Latency | 512 | 1 | 10 |
| | Training Peak Memory | 512 | 1 | 10 |

## J. Comparison against Sparse Adapters

Table 11 presents additional accuracy results of fine-tuned Llama models on the commonsense reasoning tasks using SketchTune and sparsity based PEFT methods, including SpIEL (Ansell et al., 2024) and SMT (He et al., 2025). For SketchTune, we use the INT4 data representation and GPR=4 for model sketching, while baseline methods use the original weights. SketchTune is able to achieve better or comparable accuracy consistently across different tasks while using

$2.71 - 3.54 \times$ smaller base models.

Table 11. Accuracy of SketchTune compared to sprasity-based PEFT methods for fine-tuning Llama models on commonsense reasoning datasets. Baseline results are taken from He et al. (2025). SketchTune achieve better or comparable accuracy while using sketched models that are smaller than the full base models used by baseline methods.

| Model | Method | Base Model (GB) | Trainable Param(M) | BoolQ | PIQA | SIQA | HellaSwag | Wino | ARC-e | ARC-c | OBQA | Avg. |
|---|---|---|---|---|---|---|---|---|---|---|---|---|
| LLaMA-7B | SpIEL | 13.48 | 56.6 | 67.7 | 81.2 | 78.6 | 84 | 80.2 | 78.3 | 62.8 | 78.8 | 76.5 |
| | SMT(Best) | 13.48 | 330.9 | 72 | 82.9 | **80.7** | 93.3 | 82.4 | 86.1 | 70.6 | 83 | 81.4 |
| | SketchTune$_{GPR=4}$ | 4.02 | 87.0 | **72.1** | **85.6** | 80.2 | **93.7** | **84.6** | **86.2** | **71.0** | **84.8** | **82.3** |
| LLaMA-13B | SpIEL | 26.03 | 45.8 | 73.2 | 84.3 | 81.4 | 91.2 | 84.1 | 83.1 | 68.8 | 82.8 | 81.1 |
| | SMT(Best) | 26.03 | 330.9 | 72.6 | 86.1 | 81.9 | 95 | 86.1 | 88.2 | **77.1** | 87.4 | 84.3 |
| | SketchTune$_{GPR=4}$ | 7.36 | 136.3 | **73.9** | **87.4** | **82.5** | **95.6** | **86.1** | **90.3** | 75.7 | **89.4** | **85.1** |
| LLaMA-2-7B | SpIEL | 13.48 | 55.9 | 70.5 | 80.6 | 80.8 | 85.8 | 83.4 | 81.2 | 65.8 | 81.8 | 78.3 |
| | SMT(Best) | 13.48 | 330.9 | 72.6 | 85.2 | **82** | **94.4** | **85.7** | **87.8** | 74.5 | 85 | 83.4 |
| | SketchTune$_{GPR=4}$ | 4.05 | 87.0 | **73.3** | **86.2** | 81.2 | 94.1 | 85.4 | 87.6 | **75.2** | **85.8** | **83.6** |
| LLaMA-3-8B | SpIEL | 16.06 | 47.2 | 72.1 | 83.6 | 80 | 91.8 | 85.4 | 91.2 | 76.8 | 80.8 | 82.7 |
| | SMT(Best) | 16.06 | 202.8 | **75.1** | 89.9 | 82.4 | **96.3** | **88.8** | 92.6 | **82.8** | **89.6** | **87.2** |
| | SketchTune$_{GPR=4}$ | 5.92 | 88.1 | 75.0 | **90.2** | **82.7** | 95.9 | 88.2 | **92.6** | 82.1 | 89.4 | 87.0 |

# K. LLM-as-a-judge Comparison with LoftQ

Table 12. LLM-as-a-judge evaluation between SketchTune and LoftQ

| Method | Data Type | Win | Loss | Tie | Win Rate | Loss Rate | Tie Rate |
|---|---|---|---|---|---|---|---|
| LoftQ$_{rank=64}$ | NF4 | 93 | 147 | 60 | 0.31 | 0.49 | 0.39 |
| SketchTune$_{GPR=8}$ | INT4 | 147 | 93 | 60 | 0.49 | 0.31 | 0.61 |

In this section, we evaluate the performance of SketchTune in comparison to LoftQ on language generation tasks using the LLM-as-a-judge framework (Zheng et al., 2023). Both methods were fine-tuned on the Llama-3-8B model (Dubey et al., 2024) using 4,096 randomly selected inputs from the Alpaca dataset (Taori et al., 2023). The fine-tuned models then generated responses on 300 distinct test inputs, which were also randomly sampled from the same dataset. GPT-4o was used as a judge to evaluate the models' generation quality. As illustrated in Table 12, SketchTune achieved a win-loss ratio of 0.61 against LoftQ, demonstrating superior language generation capabilities.

# L. Math Evaluations: Regular Expression Matching vs. LLM-Based Judging

For the experimental results in Table 1, we adopt the evaluation approach used in prior works (Hu et al., 2023; Yang et al., 2024b), where the last number in an LLM's response is extracted via regular expression matching and treated as the predicted answer. For instance, if an LLM outputs "Thus, Alice would need 570 tiles to cover 36 sqft area.", the method extracts 36 as the answer. However, this extraction is incorrect, as the intended answer is 570, but the regex-based approach mistakenly identifies 36.

To address this issue, we propose a more reliable evaluation method using an LLM as the answer extractor. Specifically, we use o3-mini to extract the final answer from each response. We observe that this LLM-based judging improves accuracy on math datasets by several percentage points compared to regular expression matching. Table 13 presents a detailed comparison of results for LoRA, DoRA, and our proposed method, SketchTune.

*Table 13.* Accuracy score comparison between LLM-based judgement and regex-based extraction.

| Dataset | Eval Method | LoRA$_{r=2}$ | LoRA$_{r=4}$ | LoRA$_{r=8}$ | LoRA$_{r=16}$ | DoRA$_{r=2}$ | DoRA$_{r=4}$ | DoRA$_{r=8}$ | DoRA$_{r=16}$ | Sketch-Tune$_{GPR=4}$ | Sketch-Tune$_{GPR=8}$ |
|---|---|---|---|---|---|---|---|---|---|---|---|
| AQuA | Regex | 28.3 | 28.7 | 25.2 | 24.4 | 25.6 | 26.8 | 26.0 | 25.6 | 28.7 | 29.1 |
| | o3-mini | **39.8** | **34.6** | **39.0** | **38.6** | **35.0** | **35.4** | **38.2** | **37.4** | **37.0** | **39.0** |
| GMS8K | Regex | 66.9 | 66.3 | 69.2 | 68.8 | 66.3 | 67.2 | 68.8 | 69.4 | 68.2 | 68.8 |
| | o3-mini | **68.8** | **69.4** | **71.0** | **71.0** | **68.3** | **69.4** | **70.9** | **72.0** | **71.6** | **71.4** |

