# OpenReview forum: "Sketch to Adapt: Fine-Tunable Sketches for Efficient LLM Adaptation"
_ICML.cc/2025/Conference — ICML 2025 poster_

### Official Review · Reviewer_kWKb · 2025-03-13

**Overall Recommendation:** 3

**Summary:**

This paper is concerned with parameter-efficient fine-tuning. It is argued in this paper that previous solutions in parameter-efficient fine-tuning are either low-rank or quantized. They are limited in applicability due to restrictive assumptions. In this paper, a sketchtune approach is proposed with the inspiration of sketching. By rigorously experimenting with LLaMA-1/2/3 models on math problem-solving, commonsense reasoning, and instruction-following tasks, it is shown that the proposed sketchtune is way better than considered baselines ranging from LoRA, DoRA, to S2FT in terms of both model performance and parameter efficiency. Specifically, sketchtune can achieve the same performance with baselines with smaller base models and fewer trainable parameters.

**Claims And Evidence:**

The claims in the submission are supported by clear and convincing evidence.

**Essential References Not Discussed:**

N/A

**Experimental Designs Or Analyses:**

The experimental designs and analyses are sound and adequate.

**Methods And Evaluation Criteria:**

The proposed methods are mostly elaborated and the evaluation criteria is well justified. However, I still have several concerns:
1) The sketch strategy is somehow hard to understand, specifically, in section 2.3, it is not very clear how the mapping matrix, i.e. M, is determined.

**Other Comments Or Suggestions:**

N/A

**Other Strengths And Weaknesses:**

Weaknesses:
1) Equation 6 is mis-leading, the formulation of derivative seems upside-down.
2) The dedicated CUDA kernel would better be attached as supplementary material.

**Questions For Authors:**

N/A

**Relation To Broader Scientific Literature:**

The contributions are built upon previous studies in parameter-efficient fine-tuning methods, e.g. LoRA and QLoRA. LoRA and QLoRA-like methods are limited in that they are either low-rank or quantized. So sketchtune proposes optimizations toward the mentioned limitations.

**Theoretical Claims:**

The proofs for theoretical claims are correctly provided.

---

> ### Author Rebuttal · Authors · 2025-04-01
>
> We thank the reviewer for recognizing the soundness of our work and providing thoughtful feedback. We address the reviewer’s concerns and suggestions below:
>
> ## **[Concern 1 - The sketch strategy is somehow hard to understand. Specifically, in Section 2.3, it is not very clear how the mapping matrix, i.e., M, is determined.]**
>
> Thank you for pointing this out. We provide further clarification below on how the mapping matrix $M$ is learned.
>
> Each column of $M$ is a binary one-hot vector that maps an original parameter (from a row of size $c$) to one of the $k$ entries in the sketched parameter vector, where $k \ll c$. This mapping inevitably introduces some error in the model output. To minimize this error, we learn the columns of $M$ sequentially and iteratively update the remaining unmapped parameters to compensate for the introduced error after each step.
>
> Concretely, for each original parameter, we identify the entry in the sketched parameter vector that is closest to it and assign the corresponding one-hot value in $M$. After fixing this column of $M$, we apply an update $\boldsymbol\delta$ (as defined in Equation 13) to the remaining unmapped parameters to absorb the approximation error. This process is repeated until all columns of $M$ are assigned.
>
> We will clarify this procedure in the final version of the paper.
>
> ## **[W1 - Misleading Equation 6]**
>
> We thank the reviewer for pointing out this typo. We will correct this in the final paper.
>
> ## **[W2 - Release CUDA kernel as supplementary material]**
>
> Thank you for the great suggestion. We will include the CUDA kernel in the appendix and release our code and models for the final version.

---

### Official Review · Reviewer_5Qcf · 2025-03-14

**Overall Recommendation:** 4

**Summary:**

The paper introduces  a method that compresses pre-trained LLM weights row-by-row into a smaller, shared set of trainable “sketched” weights. They compress weights by approximately minimizing the reconstruction error for activations over a set of data. Experimentally, SketchTune shows advantages in terms of model size and efficient inference with minimal performance trade-offs.

**Claims And Evidence:**

Main Idea / Theoretical Analysis: SketchTune replaces a low-rank update assumption with a sketch-based approximation, where theoretical arguments (Section 3) show that if weight updates are not actually low-rank, a sketch can give a better approximation. Under some assumptions, the paper characterizes conditions (power-law decay of singular values) under which sketches outperform low-rank approximations.
Empirical Results: On math (e.g., GSM8K) and commonsense tasks, SketchTune achieves performance on par with or surpassing strong PEFT baselines (LoRA, DoRA, etc.), even though it uses a smaller “sketched” base model. The method also compares favorably to quantized PEFT approaches like LoftQ.

However, one notable methodological detail is that SketchTune uses additional C4 data to compute the Hessian-based weighting for the sketch, effectively a calibration pass, whereas competing methods do not use additional data or have a previous data dependent compression step. Also, Table 9 in the appendix shows this “sketching” step can be a significant overhead relative to normal training times, as it involves an extra pass over substantial data and computes second-order derivatives.

**Essential References Not Discussed:**

The coverage of low-rank, quantization, and other compressive approaches is fairly thorough. However, methods focusing purely on model sparsity as an alternative – e.g. forcing adapter updates to remain in a sparse subset – only receive brief mention. A more direct empirical comparison would strengthen the position of SketchTune among all “competing” approaches.

**Experimental Designs Or Analyses:**

The experiments are generally strong, spanning multiple tasks and baselines (low-rank, quantized). Performance improvements hold across tasks. The main concern is the use of additional data and overhead of sketch computation (Appendix I), which can be quite large compared to normal finetuning time. Specifically, Table 9 suggests that multi-hour overhead might be incurred on some GPUs. This overhead may offset the claimed efficiency advantages in certain settings. I would also like to see more quantization (or actually quantization+LoRA) baselines, especially data dependent quantization methods.

**Methods And Evaluation Criteria:**

The paper evaluates on standard math, commonsense, and language modeling tasks, reporting the usual metrics. The authors also benchmark training and inference time (time to first token, decoding latency), along with memory usage. These criteria align well with the stated aim of PEFT.

**Other Comments Or Suggestions:**

NA

**Other Strengths And Weaknesses:**

Strengths:
Principled approach that is complemetary to low-rank updates.
Strong empirical results vs. both low-rank adapters and quantized finetuning.
Dedicated GPU kernel designs.

Weaknesses:
Requires a calibration pass on data to compute Hessians and sketch the pre-trained model, which can add substantial overhead.
Sparse adapters are not thoroughly compared.

**Questions For Authors:**

Time/Compute Overhead: Have you explored sampling a smaller subset of C4 or using fewer Hessian blocks to reduce overhead, and what are the trade-offs in final accuracy?
Sparse Adapters: Could row-wise sketches be combined or compared directly with a learned sparse subset of weights (like Diff-Pruning)?
Dynamic Remapping: Would re-mapping columns or recalculating the Hessian mid-training yield significantly better final performance, or do diminishing returns make this impractical?

**Relation To Broader Scientific Literature:**

SketchTune builds on prior compressive adaptation ideas (random hashing, count-sketch) and extends them with a layerwise, Hessian-based weighting. It competes directly with low-rank (LoRA, DoRA) and quantization-based (LoftQ, QLoRA) methods but has limited discussion or comparison with pure sparsity-based finetuning.

**Theoretical Claims:**

I didn't check it thoroughly, but it seems to be correct. I am not sure how mild is the assumption that the random mapping is uncorrelated with the true update matrix.

---

> ### Author Rebuttal · Authors · 2025-04-01
>
> Thank you for the thorough review and thoughtful feedback. Below, we address your questions and concerns.
>
> ## **[Theoretical Claims]**
>
> The effectiveness of LoRA and SketchTune depends on the structure of the true update matrix $\Delta$. If $\Delta$ is low-rank, LoRA is favored; if $\Delta$ aligns with the mapping matrix, SketchTune performs better.
>
> Theorem 3.1 addresses how to choose between methods without detailed knowledge of $\Delta$. Assuming only the singular value distribution, the most neutral case is when $\Delta$ is uncorrelated with the mapping. Under this assumption, Theorem 3.1 shows that as $\Delta$ becomes less low-rank, SketchTune is increasingly likely to outperform LoRA.
>
> We adopt the uncorrelated setting in Theorem 3.1 to avoid bias toward either method. Empirically, however, our analysis (Figure 1) shows that $\Delta$ is correlated with the mapping in a way that favors SketchTune.
>
> ## **[Experimental Designs 1 - More Data-Dependent Quantization + LoRA Baselines]**
>
> We performed additional comparison with data-dependent AQLM [4] quantization + LoRA baselines. We fine-tuned LoRA adapters with a 2-bit AQLM Llama-2-7B model on GSM8K using a learning rate of 1e-5 and a batch size of 16 for 5 epochs. We report the results in the table below. We are actively trying other hyperparameters to improve performance.
>
> |Model|Method|Trainable Param (M)|GSM8K|
> |-|-|-|-|
> |Llama-2-7B|$\text{AQLM}_{\text{2-bit}}$|159.91|2.81|
> | |$\text{SketchTune}_{GPR=4}$|21.75|**29.95**|
>
> ## **[W1 – Additional Data for Calibration & Sketching Overhead]**
>
> The sketching step is efficient, requires only a small calibration dataset, and is performed once per base model. Specifically, we use just 128 sequences of 2048 tokens, which is minimal compared to typical fine-tuning datasets. For further details and results, we kindly refer you to our response to Reviewer WZr9 under **[W1]**.
>
> ## **[W2 - Sparse adapters are not thoroughly compared]**
>
> Tables 1 and 2 in our paper include comparisons with a sparsity-based method, S2FT. Additionally, while we aimed to compare with Diff-Pruning[1], we encountered a challenge: the official implementation does not support LLMs such as Llama. In the table below, we instead provide additional comparison with SpIEL[2] and SMT[3], two recent sparsity-based PEFT methods, reporting average accuracy across eight commonsense reasoning tasks.
>
> |Model|Method|Base Model (GB)|Trainable Param (M)|Avg Acc.|
> |-|-|-|-|-|
> | Llama-2-7B|SpIEL|13.48|55.9|78.3|
> | |SMT(Best)|13.48|330.9|83.4|
> | |$\text{SketchTune}_{GPR=4}$|4.05|87.0|**83.6**|
> | Llama-3-8B|SpIEL|16.06|47.2|82.7|
> | |SMT(Best)|16.06|202.8|**87.2**|
> | |$\text{SketchTune}_{GPR=4}$|5.92|88.1|87.0|
>
> ## **[Q1 - Time/Compute Overhead]**
>
> We ran additional experiments to assess the impact of using smaller C4 subsets for calibration, evaluating model quality via WikiText2 perplexity and MMLU accuracy.
>
> As shown in the table below, larger calibration sets improve model quality but **do not significantly increase sketching time**, since the dominant cost in sketching comes from clustering to obtain the $k$ centers (Equation 4).
>
> These results highlight a trade-off: even a single sequence yields a usable sketch, but larger sets lead to better performance.
>
> |Model|# of Sequences|Sketch Time (min)|WikiText2 Perplexity|MMLU Accuracy|
> |-|-|-|-|-|
> |Llama-3-8B|128|39.88|6.52|60.56|
> | |32|36.62|6.59|60.36|
> | |8|32.88|6.65|60.40|
> | |1|31.70|7.00|58.58|
>
> ## **[Q2 - Sparse Adapters]**
>
> Yes, SketchTune can be combined with Diff-Pruning for greater parameter efficiency. This involves keeping the sketched parameters frozen and learning a sparse, task-specific update $\delta_\tau$, so that $w_{sketched, \tau} = w_{sketched} + \delta_\tau$.
>
> Following Diff-Pruning, $\delta_\tau$ can be modeled as a gated update using a Hard-Concrete distribution to approximate a binary mask $z_\tau$, with $\delta_\tau = z_\tau \odot w_\tau$, where $w_\tau$ is the dense task-specific update.
>
> Implementing this extension is non-trivial, and we defer it to future work.
>
> ## **[Q3 - Dynamic Remapping]**
>
> Since each sketched parameter is shared by multiple model parameters, re-mapping columns or recalculating the Hessian mid-training may help better align the sketch with the evolving gradients, potentially leading to improved final performance. Yet such an approach may introduce non-trivial computational overhead, as it would require additional clustering steps and second-order estimation during training. We defer a thorough investigation of dynamic remapping strategies to future work.
>
> **References**
> 1. Parameter-efficient transfer learning with diff pruning
> 2. Scaling sparse fine-tuning to large language models
> 3. Sparse Matrix in Large Language Model Fine-tuning
> 4. Extreme compression of large language models via additive quantization

---

### Official Review · Reviewer_v96N · 2025-03-18

**Overall Recommendation:** 4

**Summary:**

The paper proposes SketchTune, which uses a learned sketching algorithm to compress the LLM into a small set of shared sketched parameters and fine-tune those parameters for adaptation. The proposed approach reduces model size while preserving the pre-trained capabilities of the full model.

## update after rebuttal
I appreciate the authors’ efforts in providing a response, and most of my concerns have been addressed. Accordingly, I will increase my score.

**Claims And Evidence:**

Yes, the paper provides extensive experiments and theoretical analysis to support its claims.

**Essential References Not Discussed:**

I think the authors included the necessary references to support their claims.

**Experimental Designs Or Analyses:**

In LLM fine-tuning, instruction-following benchmark results are typically included (e.g., the Alpaca GPT-4 dataset with MT-Bench scores; refer to the S2FT paper as an example). Given the significant potential of the proposed method as an efficient LLM compression and fine-tuning technique, I am curious whether it performs well on instruction-following benchmarks. Additionally, why do the authors present GPR=4 only in Table 2, while Table 1 includes GPR=1, 2, 4, and 8?

**Methods And Evaluation Criteria:**

Yes, the proposed methods and evaluation criteria are reasonable.

**Other Comments Or Suggestions:**

What is the overhead of sketching time in Table 9 of the Appendix? It would be helpful to know how much memory and computation are required for learning to sketch LLM weights (described in Algorithm 1).

**Other Strengths And Weaknesses:**

The authors develop a custom CUDA kernel optimized for the specific operations required by SketchTune.

**Questions For Authors:**

Please refer to my questions for authors in the previous sections.

**Relation To Broader Scientific Literature:**

This paper introduces a new research direction that overcomes the limitations of existing PEFT methods based on low-rank or sparse assumptions.

**Theoretical Claims:**

I find Section 2.1 to be slightly misleading, as its title, "Motivation: Weight Updates Are Far from Low-Rank," suggests that the section will primarily discuss the fundamental characteristics of weight updates and why they are far from low-rank. However, the text mainly focuses on empirically comparing the effectiveness of low-rank matrices and sketching techniques in terms of approximation error, which seems more like an outcome rather than a motivation. Since the authors prove that SketchTune is well-suited to approximate delta when eta is close to 0 and delta is nearly full-rank in Section 3, providing a more in-depth analysis of empirical observations on whether weight updates of LLMs are actually close to full-rank would better support the authors' theoretical claims. This would also clearly establish the motivation for why weight updates are far from low-rank.

---

> ### Author Rebuttal · Authors · 2025-04-01
>
> We thank the reviewer for the detailed review and insightful feedback. We address your concerns as follows.
>
> ## **[Theoretical Claims 1 - Ambiguous Title in Section 2.1 & More in-depth Empirical Observation Analysis]**
>
> We thank the reviewer for the insightful suggestion. We conducted additional analysis to examine whether the weight updates in LLMs are close to full rank. In the table below, we report **the minimum rank and the standard deviation**, across all layers, required to explain a given percentage of the variance in the weight updates. This was computed by performing SVD on each weight update and counting how many top singular values are needed to reach the target variance threshold.
>
> The results show that the weight updates are relatively **high-rank**. This suggests that standard LoRA configurations (e.g., rank 32) capture only a limited portion of the variance, and may not effectively approximate the full weight updates. We will incorporate this analysis into the final version of the paper to better align with the section title.
>
> | Model                                | Max Rank      | 25% Variance        | 50% Variance         | 75% Variance         | 90% Variance         | 95% Variance         |
> |--------------------------------------|---------------|----------------------|------------------------|------------------------|------------------------|------------------------|
> | Llama 2 7B, fine-tuned on Vicuna     | 4096          | 140.4 ± 62.7         | 460.9 ± 178.2          | 1089.0 ± 358.5         | 1866.4 ± 520.5         | 2354.0 ± 581.4         |
> | Llama 3 8B, fine-tuned on OpenChat   | 1024 or 4096  | 197.5 ± 111.8        | 562.5 ± 304.7          | 1180.5 ± 607.7         | 1848.8 ± 903.9         | 2221.8 ± 1051.3        |
> | Qwen 2.5 7B, fine-tuned on Code      | 512 or 3584   | 235.8 ± 164.1        | 594.9 ± 385.6          | 1138.2 ± 691.5         | 1675.3 ± 960.5         | 1958.1 ± 1085.4        |
>
> ## **[Experimental Designs 1 – Additional Instruction-Following Benchmark]**
> We conducted additional experiments by instruction-tuning Mistral-7B on the Alpaca-GPT4 dataset for one epoch, following the experimental setup in S2FT. In the table below, we report MT-Bench scores evaluated by GPT-4, with baseline results taken from the S2FT paper. Despite using a smaller base model, SketchTune outperforms the baselines.
>
> | Model | Method | Base Model (GB) | Writing | Roleplay | Reasoning | Code | Math | Extraction | STEM | Humanities | Avg. |
> |---|---|---|---|---|---|---|---|---|---|---|---|
> | Mistral-7B | Full FT | 14.48 | 5.50 | 4.45 | 5.45 | 2.50 | 3.25 | 5.78 | 4.75 | 5.45 | 4.64 |
> |  | $\text{S}^2\text{FT}$ | 14.48 | 6.95 | 4.40 | 5.50 | 2.70 | 3.55 | 5.95 | 6.35 | 6.75 | 5.27 |
> |  | $\text{SketchTune}_{\text{GPR}=4}$ | 4.66 | 4.60 | 5.20 | 9.23 | 3.05 | 4.80 | 7.45 | 8.13 | 8.45 | **6.36** |
>
> ##  **[Experimental Designs 2 – GPR Choice in Tables 1 & 2]**
> We reported results for GPR=4 in Table 2 due to the time-consuming nature of fine-tuning on the Commonsense170K dataset, which is 17× larger than the Math10K dataset used in Table 1. A single training run takes several days to complete. We selected GPR=4 as a practical trade-off between memory efficiency and final performance. We are happy to conduct additional experiments with other GPR values and include the results in the camera-ready version.
>
> ## **[Comments 1 – Overhead of Sketching Time and Memory Requirement]**
>
> In Table 9 of the paper, the reported sketching time refers to the time required to compress a base model into a sketched model using a single RTX 8000-48GB GPU. In the table below, we provide additional sketching time (INT4, GPR=4) and peak GPU memory usage for models of various sizes, using a single A100-40GB GPU. Thanks to the layer-wise optimization objective (Equation 1), model sketching scales efficiently to large models (e.g., 70B) on a single GPU. Additionally, we plan to upload sketched models to the HuggingFace Model Hub, enabling users to directly download and fine-tune without repeating the sketching step.
>
> | Model         | Original Size | Sketched Size | Max GPU Memory | Sketching Time |
> | ------------- | ------------- | ------------- | --------------- | --------------- |
> | Llama-3.2-3B  | 6.43G         | 3.18G         | 9.92 GB         | 20.7 mins       |
> | Llama-3.1-8B  | 16.07G        | 5.92G         | 18.37 GB        | 41.62 mins      |
> | Llama-3.1-70B | 141.12G       | 40.15G        | 28.05 GB        | 266.87 mins     |

---

> > ### Comment · Reviewer_v96N · 2025-04-05
> >
> > I appreciate the authors’ efforts in providing a response. Regarding the overhead of sketching time and memory requirements, could you also provide the cost of fine-tuning sketches? This would help me better understand the relative cost of learning to sketch compared to fine-tuning sketches.

---

> > > ### Author Response · Authors · 2025-04-05
> > >
> > > We thank the reviewer for the thoughtful comment. To address the question, we have added a table below presenting the time cost of sketching and fine-tuning on a single A100-40GB GPU. As shown, fine-tuning accounts for the majority of the end-to-end time, especially for larger datasets like Commonsense170K. We also note that sketching is a one-time process per model. To improve usability, we will release the sketched models on the Hugging Face Model Hub, allowing users to directly download them and bypass the sketching step.
> > >
> > > | Model | Dataset | Epochs | Sketching Time | Fine-tuning Time | Sketching Time / Total Time |
> > > |---|---|---|---|---|---|
> > > | Llama 3 8B (INT4, GPR=4) | Commonsense170K | 2 | 41.62 mins | 1244.43 mins | 3.24% |
> > > | Llama 3 8B (INT4, GPR=4) | Math10K | 4 | 41.62 mins | 160.17 mins | 20.60% |

---

### Official Review · Reviewer_WZr9 · 2025-03-19

**Overall Recommendation:** 4

**Summary:**

The paper proposes an alternative to parameter efficient fine-tuning of LLMs by using sketching to create a low-dimensional representation of the weight matrices which is theoretically shown to be better for certain classes of matrices. Experiments on Llama models shows that the algorithm is able to outperform PEFT while using smaller base models and comparable trainable parameters.

## update after rebuttal

The authors have addressed my concerns by adding experiments showing that the overhead of sketch generation is acceptable and clarifying that their choice of hyperparameters allows for significant compression despite grouping. Therefore, I have increased my score.

**Claims And Evidence:**

Yes

**Essential References Not Discussed:**

N/A

**Experimental Designs Or Analyses:**

I checked the experiments in the main paper, and they appear correct to me.

**Methods And Evaluation Criteria:**

Yes

**Other Comments Or Suggestions:**

See questions below

**Other Strengths And Weaknesses:**

N/A

**Questions For Authors:**

The derivation of (6) is not clear to me. Please explain how it is derived.

**Relation To Broader Scientific Literature:**

Strengths:

1. Sketching significantly reduces the model size while preserving the pre-trained capabilities of the model.

2. Experiments show that it achieves higher accuracy than PEFT baselines with comparable number of trainable parameters.

Weaknesses:

1. The process of generating the sketches appears to be more expensive than LoRA

2. It seems like learning separate c-dimensional sketches for each of the 'g' subgroups in a k-dimensional row can reduce the memory efficiency since we need kg << c to get significant memory saving and that might be difficult to satisfy

**Theoretical Claims:**

I checked the proofs at a high-level and they appear correct to me.

---

> ### Author Rebuttal · Authors · 2025-04-01
>
> We thank the reviewer for their thoughtful feedback and for acknowledging the empirical effectiveness of our method. We address your concerns and questions below:
>
> ## **[W1 - Sketch generation process appears to be more expensive than LoRA]**
>
> While SketchTune introduces an additional sketching step before fine-tuning, this preprocessing is **fast, resource-efficient, and one-time** per base model. In the table below, we report additional end-to-end sketching time (INT4, GPR=4) for different sized models, using a single A100-40GB GPU. Thanks to our layer-wise optimization objective (Equation 1), sketching scales efficiently to large models (**70B**) with a single GPU.
>
> | Model         | Original Size | Sketched Size | Max GPU Memory | Sketching Time |
> | ------------- | ------------- | ------------- | --------------- | --------------- |
> | Llama-3.2-3B  | 6.43G         | 3.18G         | 9.92 GB         | 20.7 mins       |
> | Llama-3.1-8B  | 16.07G        | 5.92G         | 18.37 GB        | 41.62 mins      |
> | Llama-3.1-70B | 141.12G       | 40.15G        | 28.05 GB        | 266.87 mins     |
>
> The sketching overhead is comparable to existing compressed PEFT baselines. For example, LoftQ [1] reports a quantization overhead of **21 seconds** for a 4096 $\times$ 4096 matrix, while SketchTune’s overhead is only **5 seconds (4.2x speedup)**.
>
> Moreover, a single sketched model can be reused across multiple downstream tasks. As demonstrated in Tables 1 and 2 of the paper, the same sketched models are effective across different domains, i.e. math and commonsense reasoning. We also plan to release these models on the HuggingFace Model Hub, enabling users to directly download and fine-tune without repeating the sketching step.
>
>
> ## **[W2 - Learning separate sketches for each of the $g$ subgroups may reduce memory efficiency]**
>
> Our sketching approach is memory-efficient and effectively compresses the model weights. Specifically, each row $\mathbf{w} \in \mathbb{R}^{1 \times c}$ is divided into $g$ non-overlapping sub-rows $\mathbf{w}' \in \mathbb{R}^{1 \times \frac{c}{g}}$, each of which is sketched into a $k$-dimensional vector. The resulting sketched row $\mathbf{w}_{\text{sketched}} \in \mathbb{R}^{1 \times gk}$ leads to a compression factor of $\frac{c}{gk}$.
>
> For Llama 2 7B, the row size $c$ is either 4096 or 11008. In our experiments, we set $k \in \\{4, 8, 16\\}$ and $g \in \\{1, 2, 4, 8\\}$, resulting in a compression factor of at least $32\times$ for Llama 2 7B. Notably, the compression factor increases with larger models, due to larger $c$ and fixed $k$ and $g$.
>
> Thus, even with subgrouping, the sketching approach remains highly memory-efficient across model scales.
>
>
> ## **[Q1 - The derivation of (6) is not clear to me. Please explain how it is derived.]**
>
> Thank you for pointing this out. We realize that Equation (6) contains a typo: the derivative fractions are upside down. We apologize for the oversight and will correct this in the camera-ready version. The correct expression is:
>
> $\frac{\partial \mathcal{L}}{\partial w_{\text{sketched}}} = \frac{\partial \mathcal{L}}{\partial y} (MX)^\top$
>
> The equation is derived as follows. Let $X$ be the layer input, $w_{\text{sketched}}$ the sketched row, and $M$ the mapping matrix. The forward pass computes:
>
> $y = w_{\text{sketched}}MX$
>
> During backpropagation, we apply the chain rule to compute the gradient of the loss $\mathcal{L}$ with respect to $w_{\text{sketched}}$:
>
> $\frac{\partial \mathcal{L}}{\partial w_{\text{sketched}}} = \frac{\partial \mathcal{L}}{\partial y} \cdot \frac{\partial y}{\partial w_{\text{sketched}}} = \frac{\partial \mathcal{L}}{\partial y} (MX)^\top$
>
> We will clarify this derivation in the revised paper.
>
> **References**
>
> [1] Li, Yixiao, et al. "LoftQ: LoRA-Fine-Tuning-aware Quantization for Large Language Models." The Twelfth International Conference on Learning Representations.

---

> > ### Comment · Reviewer_WZr9 · 2025-04-09
> >
> > Thank you for addressing all my concerns. I have increased my score.

---

### Decision · Program_Chairs · 2025-05-01

**Decision:**

Accept (poster)

**Comment:**

This paper introduces SketchTune, a novel parameter-efficient fine-tuning (PEFT) technique based on learning sketches of model weights. This approach provides an alternative to prevalent low-rank adaptation (LoRA) and quantization methods. The reviewers generally found the method well-motivated and the empirical results strong, demonstrating superior performance compared to several established PEFT baselines across various tasks (math, commonsense reasoning, instruction-following) while often using smaller base models or fewer trainable parameters. The authors successfully addressed most of the reviewers' concerns during the rebuttal phase, leading two reviewers (WZr9, v96N) to raise their scores.

**Consolidated Strong Points:**

* **Novelty and Strong Empirical Performance (All Reviewers):** SketchTune offers a principled alternative to existing PEFT methods, backed by strong empirical results showing consistent improvements over LoRA, DoRA, S2FT, LoftQ, and added sparsity baselines (SpIEL, SMT) on diverse benchmarks.
* **Parameter and Computational Efficiency (WZr9, v96N, 5Qcf):** The method achieves significant model compression and demonstrates efficiency benefits, supported by a custom CUDA kernel. It often matches baseline performance with smaller underlying models.
* **Theoretical Motivation (WZr9, v96N, 5Qcf):** The paper provides theoretical grounding, arguing SketchTune is particularly advantageous when weight updates deviate from low-rank structure, a claim supported by additional empirical analysis provided in the rebuttal.
* **Responsiveness to Feedback (All Reviewers):** Authors provided detailed clarifications, corrected minor issues, added requested experiments (instruction-following, new baselines), and offered further analysis during rebuttal.

**Consolidated Concerns (and how they were addressed):**

* **Initial Sketching Overhead (WZr9, v96N, 5Qcf):** Concerns about the one-time cost (time, compute, calibration data) of the sketching step. Addressed by authors providing benchmarks demonstrating manageable cost, clarifying minimal data requirements, showing cost/performance trade-offs, and promising pre-sketched model release. *Reviewers WZr9, v96N, 5Qcf accepted the response.*
* **Clarity and Minor Errors (WZr9, v96N, 5Qcf, kWKb):** Initial lack of clarity regarding Eq. 6 derivation, Section 2.1 motivation, mapping matrix M determination, and a theoretical assumption. Addressed via corrections, additional empirical analysis, detailed explanations, and justifications in the rebuttal. *All relevant reviewers acknowledged the clarifications.*
* **Initial Experimental Gaps (v96N, 5Qcf):** Lack of instruction-following results and limited comparison to certain quantization/sparsity methods. Addressed by adding MT-Bench results and comparisons to AQLM+LoRA, SpIEL, SMT. *Reviewers v96N, 5Qcf accepted the additions.*
* **Code/Kernel Availability (kWKb):** Request for the custom CUDA kernel. Addressed by authors promising code, model, and kernel release. *Reviewer kWKb acknowledged.*

Overall, the paper presents a compelling new PEFT method with strong results. While the initial sketching step adds a layer of complexity compared to methods like LoRA, the authors effectively argued its feasibility and benefits. The concerns raised were substantially addressed.